# Levels of Amyloid Beta (*Aβ*) Expression in the *Caenorhabditis elegans* Neurons Influence the Onset and Severity of Neuronally Mediated Phenotypes

**DOI:** 10.3390/cells13181598

**Published:** 2024-09-23

**Authors:** Neha Sirwani, Shannon M. Hedtke, Kirsten Grant, Gawain McColl, Warwick N. Grant

**Affiliations:** 1Department of Environment and Genetics, School of Agriculture, Biomedicine and Environment, La Trobe University, Bundoora, VIC 3086, Australia; s.hedtke@latrobe.edu.au (S.M.H.); w.grant@latrobe.edu.au (W.N.G.); 2Florey Institute of Neuroscience and Mental Health, The University of Melbourne, Parkville, VIC 3052, Australia; gawain.mccoll@florey.edu.au

**Keywords:** amyloid β, *C. elegans*, neurodegeneration, Alzheimer’s disease, behavioural assays, transgene copy number

## Abstract

A characteristic feature of Alzheimer’s disease (AD) is the formation of neuronal extracellular senile plaques composed of aggregates of fibrillar amyloid β (*Aβ*) peptides, with the *Aβ1-42* peptide being the most abundant species. These *Aβ* peptides have been proposed to contribute to the pathophysiology of the disease; however, there are few tools available to test this hypothesis directly. In particular, there are no data that establish a dose–response relationship between *Aβ* peptide expression level and disease. We have generated a panel of transgenic *Caenorhabditis elegans* strains expressing the human *Aβ1-42* peptide under the control of promoter regions of two pan-neuronal expressed genes, *snb-1* and *rgef-1*. Phenotypic data show strong age-related defects in motility, subtle changes in chemotaxis, reduced median and maximum lifespan, changes in health span indicators, and impaired learning. The *Aβ1-42* expression level of these strains differed as a function of promoter identity and transgene copy number, and the timing and severity of phenotypes mediated by *Aβ1-42* were strongly positively correlated with expression level. The pan-neuronal expression of varying levels of human *Aβ1-42* in a nematode model provides a new tool to investigate the in vivo toxicity of neuronal *Aβ* expression and the molecular and cellular mechanisms underlying AD progression in the absence of endogenous *Aβ* peptides. More importantly, it allows direct quantitative testing of the dose–response relationship between neuronal *Aβ* peptide expression and disease for the first time. These strains may also be used to develop screens for novel therapeutics to treat Alzheimer’s disease.

## 1. Introduction

The primary pathological hallmark of Alzheimer’s disease (AD) is the presence of extracellular senile plaques composed of aggregates of fibrillar amyloid β (*Aβ*) peptides, derived from the proteolytic processing of amyloid precursor protein (APP) [1,2]. The transmembrane protein APP is cleaved sequentially by β-secretase at the N-terminal amino acids 1 and 11, followed by cleavage by γ-secretase, to give rise to a family of *Aβ* peptides varying between 39 and 43 amino acids in length [3]. One of the most abundant *Aβ* peptides in the human AD brain is the hydrophobic *Aβ1-42* peptide [4,5]. The amyloid cascade hypothesis suggests that the progressive and abnormal accumulation and aggregation of the *Aβ* peptides in the human AD brain is the primary cause of the disease, and this precedes the other disease symptoms, such as formation of intracellular neurofibrillary tangles, cognitive decline, and dementia [6,7,8]. However, the molecular and cellular mechanisms underlying AD pathology remain contentious and unclear [9,10,11].

Despite genetic evidence and the demonstrated involvement of *Aβ* in inducing synaptic dysfunction, disrupting neural connectivity, and association with neuronal death in a brain region-specific manner, the amount and distribution of extracellular *Aβ* deposition are only weakly correlated with the clinical expression and severity of the disease [12]. This weak correlation between the plaque burden and disease severity has been used to criticise the amyloid cascade hypothesis. Furthermore, in many cases the level of the soluble *Aβ* correlates with the disease burden better than the insoluble *Aβ* version [13,14]. Additionally, it has been suggested that the soluble *Aβ* protein is associated with faster cognitive decline, supporting the role of soluble *Aβ* as a neurotoxic agent of aging [14]. The insoluble *Aβ* appears only to signify the presence of the disease. Although in physiologic conditions and at a low concentration soluble monomeric *Aβ* may have a role in neural development and in the regulation of cholinergic transmission [15,16], it appears that *Aβ* in higher concentrations occurring as oligomers and aggregated forms causes neurotoxicity, impairs blood flow within the cerebral structure, and accelerates neuronal dysfunction [17]. Thus, the exact relationship between *Aβ* concentration and neuronal dysfunction is unclear, and more studies need to be conducted to clarify it [17].

The nematode *Caenorhabditis elegans* is an excellent in vivo model system to study the toxicity associated with these *Aβ* peptides. The simple and compact nervous system of *C. elegans* consists of 302 neurons. It has well-mapped synaptic connections determined by serial electron micrographs [18,19,20], and a number of robust behavioural phenotypes that have been used to study neuronal function such as chemotaxis, locomotion, egg-laying, pharyngeal pumping, and defecation [21,22,23,24]. The worm is not only capable of displaying a wide range of behavioural phenotypes but also shows behavioural plasticity, including learning behaviours [25].

Several *C. elegans* strains have been generated previously by expressing the *Aβ1-42* peptide in different cell types, and several behavioural phenotypes have been correlated with *Aβ1-42* expression. For instance, the *C. elegans* strain expressing *Aβ1-42* peptide in the body wall muscle cells shows a severe, fully penetrant, and age-dependent progressive paralysis phenotype [26]. Since AD is a neurodegenerative disorder, a more disease-relevant strain expressing *Aβ1-42* in the nervous system using the pan-neuronal promoter *unc-119* was reported and showed reduced lifespan and subtle age-associated decline in behavioural functions such as egg laying, pharyngeal pumping, and abnormal head movements [27]. Another study attempted to show the phenotype associated with *Aβ1-42* expression from a single inserted copy of the transgene in a single pair of glutamatergic sensory neurons, the BAG neurons. However, there was no significant behavioural phenotype reported in this strain [28]. Although there are a limited number of *C. elegans* strains expressing the relevant full-length *Aβ1-42* peptide in the neurons, expression of *Aβ* is associated with subtle behavioural deficits in these strains.

In the present study, we sought to evaluate the effects of varying levels of pan-neuronal *Aβ1-42* expression on the onset and severity of different behavioural phenotypes in worms by developing additional transgenic *C. elegans* strains of *Aβ* toxicity to achieve variation in *Aβ1-42* expression. Differences in transgene expression can be mediated by the promoters integrated with the plasmid, as well as the number of copies of the transgene. We hypothesised that the severity of the disease phenotype correlates with the levels of *Aβ* expression in these transgenic strains, assuming that transcript concentration is correlated positively with protein translation and/or accumulation. Key behaviours in *C. elegans* include lifespan, growth rate, egg laying and retention, chemotaxis, and motility. If the level of *Aβ* expression in the nervous system drives differences in disease severity, then it would be expected that as *Aβ* expression increases, so does the impact on each of the life and health span measurements as worms get older. The pan-neuronal *Aβ1-42*-expressing transgenic *C. elegans* strains described in this study showed varying levels of *Aβ1-42* expression and also showed variation in the severity of some, but not all, of the behavioural phenotypes that were measured. Overall, all the *Aβ*-expressing strains showed reduced longevity, impaired egg-laying, and age-related decline in motility in liquid media, accompanied by subtle defects in chemotaxis and some indications of deficits in neurotransmitter signalling. The phenotypic variation may be correlated with the level of *Aβ1-42* transcript abundance.

## 2. Materials and Methods

### 2.1. Plasmid Construction

The plasmid pPD49.26 was used as the backbone to construct the pan-neuronal human DA-*Aβ1-42* expression plasmids using the pan-neuronal promoters of the *snb-1* and *rgef-1* gene. The plasmid pPD49.26 contains a synthetic intron and a 3’ UTR region from the *unc-54* gene, which are essential elements required for appropriate transgene expression in the worm, as well as three polylinker (MCS) regions for cloning exogenous DNA [29]. The expression plasmid containing the human amyloid β (*Aβ*) 1-42 gene driven by the pan-neuronal promoter of the *snb-1* gene was assembled in two steps. First, the signal peptide and the *hu-DA-Aβ1-42* were extracted from the plasmid pCL354. The fragment containing the signal peptide and *hu-DA-Aβ1-42* was digested using NheI and SacI from pCL354 [26] and cloned into the MCSII of pPD49.26 to generate the promoter-less plasmid pAB42. This newly constructed promoter-less *Aβ1-42*-expressing plasmid could be used as a template for future plasmid construction, to drive the expression of *Aβ1-42* in specific subsets of neurons using different neuronal promoters. Briefly, pCL354 is an expression plasmid that consists of the peptide (*hu-DA-Aβ1-42*) driven by the muscle-specific promoter *unc-54*.

Secondly, a 2 kb region upstream of the *snb-1* gene (thought to contain the promoter–enhancer sequence) was amplified from *C. elegans* wild-type genomic DNA with *snb-1* promoter-specific forward and reverse primers that add XbaI (5’) and XmaI (3’) sites, respectively [30]. Immolase enzyme (Bioline, Inc., Cincinnati, OH, USA) was used for amplifying the *snb-1* promoter fragment in a 10 μL PCR reaction (1 μL DNA template, 0.5 μL Primer mix (10 μM), 0.4 μL dNTPs, 1 μL Buffer (10X), 0.6 μL MgCl_2_ solution (50 mM), 0.1 μL Immolase enzyme, 6.4 μL HPLC-grade water) using the following PCR conditions: denaturation at 95 °C for 10 min, 35 cycles of denaturation at 95 °C for 15 s, annealing at 61.2 °C for 40 s, and elongation at 72 °C for 2.5 min, followed by final extension at 72 °C for 5 min. The *snb-1* fragment PCR product was digested with XbaI and XmaI and cloned into the complementary XbaI (5’) and XmaI (3’) sites of MCSI of plasmid pAB42, resulting in transgene expression plasmid pSNB1AB42. As a control for appropriate pan-neuronal expression, the same *snb-1* promoter fragment was also inserted into the complementary sites XbaI (5’) and XmaI (3’) of the promoter-less GFP-containing plasmid, pPD95.77 (Fire vector kit 1995) [29], resulting in pan-neuronal GFP expression plasmid pSNB-1GFP. An additional non-toxic mouse *Aβ1-42* control strain was generated by cloning the *snb-1* promoter fragment into the MCSI of the pPD49.26 vector backbone. The mouse *Aβ1-42* transgene was extracted from the plasmid pGMC110 using the restriction enzymes NheI and SacI and cloned into the MCSII of the pPD49.26 vector containing the *snb-1* promoter fragment in the MCSI.

Next, the promoter region 2670 bp upstream of the *rgef-1* gene was amplified from the *C. elegans* wild-type genomic DNA using the *rgef-1* promoter-specific forward and reverse primers using the long-range PCR Go Taq enzyme (Promega Corporation, Madison, Wisconsin, USA) [31]. PCR was performed in 10 µL reactions (1 µL DNA template, 0.5 µL Primer mix (10 µM), 3.5 µL HPLC grade water, and 5 µL GoTaq Master mix (2X)) using the following conditions: initial denaturation at 94 °C for 2 min, 35 cycles of denaturation at 94 °C for 30 s, annealing at 69.7 °C for 30 s, and elongation at 70 °C for 3 min, followed by a final extension at 72 °C for 10 min. The amplified *rgef-1* promoter fragment was then digested with PstI and XmaI and cloned into the complementary PstI and XmaI sites of the promoter-less *Aβ* plasmid pAB42, resulting in the expression plasmid pEXAB42 containing the human *Aβ1-42* transgene driven by pan-neuronal promoter *rgef-1* (F25B3.3). The *rgef-1* promoter fragment was also inserted into the complimentary PstI and XmaI sites of the promoter-less GFP-containing plasmid pPD95.77, resulting in transgene expression plasmid pEXGFP. All clones were verified by Sanger sequencing (Macrogen, Korea). All primers used in the study were synthesised by IDT DNA Technologies (Appendix A).

### 2.2. C. elegans Maintenance and Strain Details

Standard protocols were employed for the maintenance of *C. elegans* strains [32]. All worm strains were cultured on standard nematode growth media (NGM) plates seeded with *E. coli* OP50 culture at 20 °C, unless otherwise stated [33]. The worm strains used in this study were WG731 (transgenic integrated *mCherry* control strain) (*wgIs1*[pAV1944(*myo-2p::mCherry*)]), WG643 (transgenic integrated neuronal *Aβ* strain) (*wgIs2*[pAV1944(myo*-2p::mCherry*)+ p*SNB-1*AB42 (*snb-1p::huAβ1-42*)]), WG663 (transgenic integrated neuronal *Aβ* strain) (*wgIs3*[pAV1944(*myo-2p::mCherry*) + pEXAB42(*rgef- 1p::huAβ1-42*)]), WG625 (transgenic integrated neuronal GFP strain) (*wgIs4*[pEXGFP(*rgef-1p::GFP)*]), WG700 (transgenic integrated neuronal GFP strain) (*wgIs5*[p*SNB-1*GFP(*snb-1p::GFP*)]), GRU101 (transgenic integrated YFP control strain) (*gnaIs1*[*myo-2p::YFP*]), GRU102 (transgenic integrated neuronal *Aβ* strain) (*gnaIs2*[*myo-2p::YFP* + *unc-119p::huAβ1-42*]), and CB1112 (*cat-2 (e1112)*). The standard Bristol N2 strain was used as the wild-type control.

### 2.3. Microscopy Details for Imaging the Pan-Neuronal GFP Expressing Strains

Glass microscope slides were prepared with agarose pads by pipetting 300 μL of 2% agarose in M9 buffer (22 mM KH_2_PO_4_, 42 mM Na_2_HPO_4_, 86 mM NaCl, and 1 mM MgSO_4_, sterilised by autoclaving) onto the slide and flattening slightly using a coverslip, before allowing the agarose to cool and gel. Once the agarose pad was formed, the coverslip was removed. Several worms were picked on each agarose pad and immobilised by heating the slide on a block at 55 °C for 10 s. The agarose pad was then covered with a coverslip. For imaging WG625 (*rgef-1p::GFP*) worms, images were acquired using either an Upright microscope Nikon Eclipse Ci equipped with a Nikon DS-U3 digital camera (Nikon Corporation, Tokyo, Japan) or a Zeiss LSM 510(Carl Zeiss Microscopy GmbH, Jena, Germany) laser scanning confocal microscope with a Plan Apochromatic 20X/0.8 NA objective. Excitation was achieved with an argon multiline laser at 488 nm (eGFP) and a DPSS laser of 561 nm (mCherry). A long pass LP575 filter was used for detection of mCherry. Z stacks were obtained using ZEN 2 software (Carl Zeiss Microscopy, GmbH, Jena, Germany). For imaging the WG700 (*snb-1p::GFP*) strain, an inverted Nikon Ti Eclipse microscope was used. The FITC filter was used to visualise GFP fluorescence (~495 nm) in the worms.

### 2.4. Generation of Transgenic Integrated Strains

To generate the transgenic *C. elegans* strains described in this study, pan-neuronal *hu-DA-Aβ1-42* expression plasmids were introduced into the *C. elegans* gonads by microinjection. The *Aβ*-containing plasmids pEXAB42 (*rgef-1* promoter) and p*SNB-1*AB42 (*snb-1* promoter) were micro-injected at a concentration of 25 ng/µL along with the marker (*myo-2p::mCherry)* plasmid pAV1944 (2.5 ng/μL) and sheared genomic DNA (50 ng/μL). Plasmid pAV1944 (*myo-2p::mCherry::unc-54* 3’UTR) was used as co-injection marker. pAV1944 expresses mCherry specifically in the pharyngeal muscles [34]. This plasmid was a gift from Gary Silverman at the Washington University School of Medicine in St. Louis (Addgene plasmid #37830; http://n2t.net/addgene:37830; accessed on 1 July 2015, RRIDD: Addgene_37830). The genomic DNA was digested with the 4-base cutting restriction enzyme Sau3AI by incubating the genomic DNA with 1X NEBuffer^TM^ r1.1 and 10 units of Sau3AI per microgram of DNA at 37 °C for 1 h, followed by heat inactivation of the enzyme at 65 °C for 20 min (New England Biolabs, Inc.). The progeny from the micro-injected worms were selected on the basis of the pharyngeal mCherry expression. The GFP-containing plasmids pEXGFP and p*SNB-1*GFP were microinjected along with sheared genomic DNA (50 ng/µL GFP plasmid and 50 ng/µL genomic DNA). For GFP transgenic strains, the progeny were selected based on pan-neuronal GFP expression. No marker was co-injected with p*SNB-1*GFP and pEXGFP, as the transgene itself expresses GFP, which can be easily viewed via epifluorescence microscopy. All constructs after microinjection yielded 5–7 extrachromosomal transgenic F2 lines. Of these, the line that exhibited more than 80% transmission was selected for the integration experiment.

The micro-injected worms containing the transgene were irradiated using X-rays to allow stable integration of the extrachromosomal arrays into the *C. elegans* genome. Briefly, one hundred age-synchronous L4-stage hermaphrodites were placed on an unseeded 6 cm NGM agar plate and were irradiated with X-rays (3000–4000 rad) for 4 min using an RS 2000 Biological Research Irradiator (Rad Source Technologies, Inc., Buford, GA, USA). The worms were then transferred to seeded plates and allowed to recover overnight in an incubator set at 15 °C. These X-ray-irradiated adults, also known as P0s, were then allowed to lay eggs. About 400 F1 progeny were picked individually on 6 cm seeded NGM agar plates and analysed for increased transgene transmission in the F2 generation. For an F1 that is heterozygous for an integrated array, the percentage of F2s showing the presence of an integrated array upon self-fertilisation is approximately 75%, and one third of the fluorescent F2 progeny will be homozygous for the integrated transgene array. Accordingly, about eight F2 progeny from each F1 parent showing >75% fluorescent progeny were picked individually on 6 cm NGM plates and were analysed for 100% transmission of the transgene [35]. Approximately four F3 progeny from candidate F2 homozygotes were individually picked to confirm genomic integration of the transgene and to avoid selecting sterile worms. The integrated strain was then outcrossed at least three times to the N2 strain to remove any background mutations introduced during the irradiation procedure. After integration, a total of three integrated lines were isolated for each construct.

### 2.5. Single Worm Lysis and Copy Number PCR Assay

The copy number of the integrated array was determined by quantitative PCR (qPCR) of single worms for each integrated strain.

Single-worm lysis was performed by picking individual worms into a tube containing 20 µL lysis buffer. Samples were incubated for 16 h at 55 °C followed by 1 h at 85 °C. Worm lysates diluted 10-fold with HPLC-grade water were used as a DNA template for the qPCR of all test samples.

Genomic DNA isolated from GMC296 (using the ISOLATE II Genomic DNA kit from Bioline, Inc.) was used as template DNA to generate standard curves for two PCR reactions, one for the target *Aβ1-42* gene and another for the single-copy reference gene, *Y45F10D.4*, which encodes a putative iron sulphur-containing protein [36]. Standard curves were obtained for the *Aβ1-42* gene and *Y45F10D.4* by generating a 10-fold dilution series (DNA concentration range: 1 ng to 1 × 10^−7^ ng) of the template DNA. A no-template negative control was included in all the PCR reactions.

The putative *Aβ*-containing transgenic strains, selected based on pharyngeal mCherry expression, were analysed for the *Aβ1-42* transgene copy number. Average efficiencies obtained from the standard curves were used to normalise the C_q_ value of the test samples. The relative copy number of the *Aβ1-42* gene to reference gene *Y45F10D.4* was calculated using the ΔΔC_t_ method [37].

Equation (1): Relative copy number: ΔΔC_t_ method
(1)$$Relative\;copy\;number=2^{\left[Cq\left(A\beta \right)*E\left(A\beta \right)\right]-[Cq\left(Y45F10D.4\right)*E(Y45F10D.4)}$$
where C_q_ = quantitation cycle and E = reaction efficiency.

The qPCR for each sample was performed in duplicate. The relative copy number of *Aβ1-42* in each transgenic strain was calculated by averaging the copy number of the *Aβ1-42* gene of at least eight worms for that particular transgenic strain. Since all integrated lines had comparable copy numbers, only one line per construct was used for further experiments.

The expression of the *Aβ1-42* transgene should be determined by a combination of the transgene copy number (measured by qPCR) and the intrinsic promoter activity (measured as FKPM from RNAseq data available at WormBase (www.wormbase.org; accessed on 23 June 2017). To facilitate comparisons between and interpretation of transcript abundance for the transgenic strains, we used an “index of expected expression” that was simply the arithmetic product of FKPM multiplied by the qPCR copy number.

Equation (2): Index of expected expression
(2)Index of expected expression=Aβ transgene copy number ∗ FKPM value

### 2.6. Total RNA Isolation and Quantitative Real-Time Polymerase Chain Reaction (qRT-PCR)

Total RNA was isolated from wild-type and transgenic worms using Trizol (Invitrogen, Thermo Fisher Scientific, Waltham, MA, USA). The frozen worm pellets were homogenised in 500 µL Trizol and incubated at room temperature for 10 min, to which 0.1 volumes of 1–bromo 3–chloropropane (Sigma) were added, mixed thoroughly, and incubated for another 10 min. The sample was centrifuged at 12,000× *g* for 20 min at 4 °C. The aqueous phase containing RNA was transferred to a fresh tube and 0.8 volumes of isopropanol (Sigma-Aldrich, Inc., St. Louis, MO, USA) was added. The samples were incubated at room temperature for 15 min and centrifuged at 12,000× *g* for 15 min at 4 °C. The pellet obtained was washed with 75% ethanol and centrifuged for 5 min at 7500× *g*. The pellet was then air dried and resuspended in nuclease-free water by heating at 60 °C for 10 min to dissolve the RNA. Following RNA extraction, the samples were treated with DNase I and purified (New England Biolabs, Inc., Ipswich, MA, USA) according to the manufacturer’s instructions to remove any residual genomic DNA contamination. cDNA was synthesised using the Tetro cDNA synthesis kit (Bioline, Inc., Cincinnati, OH, USA) according to the manufacturer’s instructions.

Standard curves for reference genes *cdc-42* and *Y45F10D.4* were generated using 10-fold serial dilutions of N2 cDNA and for *Aβ*, GMC296 cDNA was used as a template. RT-qPCR was performed using the same PCR reaction and conditions for qPCR as described above. The real-time quantitative RT-PCR data were normalised by geometric averaging of multiple internal control genes, as previously described, which is a modification of the Pfaffl method [38,39,40,41]. The equation used for calculating the relative gene expression using multiple reference genes is as follows:

Equation (3): Equation for calculating relative gene expression in RT-qPCR.
(3)$$Relative\;gene\;expression=\frac{{\left(E_{GOI}\right)}^{\Delta CtGOI}}{GeoMean[{\left(E_{REF}\right)}^{\Delta CtREF}]}$$

### 2.7. Lifespan Analysis

Age-synchronous worm populations were obtained by allowing young adult hermaphrodites to lay eggs for 4–6 h on 6 cm NGM plates seeded with *Escherichia coli* OP50, then carefully washing the plates to remove all worms while leaving the newly laid eggs behind. The eggs obtained were incubated at 20 °C for three days, and young worms on the first day of adulthood were used for all lifespan assays. On day 3, about 20 worms were picked on each 6 cm seeded NGM plate, for a total of 120 worms used for each assay. The number of live and dead worms were counted. The worms were transferred to fresh plates every day until the end of the reproductive period to avoid overlapping generations and to separate them from the larvae and transferred to new plates every second day during the post-reproductive period. A worm was scored as dead if there was no touch-provoked movement. In addition, the worms that were lost either by crawling off the plate (resulting in drying out) or due to internal hatching (bagging) were scored as censored and not incorporated into the analysis. The assay for each strain was repeated at least three times. Survival curves or Kaplan–Meier (KM) curves were generated and analysed in GraphPad Prism 8 software (GraphPad Software, Inc., La Jolla, CA, USA). Survival curves of the transgenic *C. elegans* strains described in Section 3.1. The Mantel–Cox (log rank) test was used to determine the difference in the distribution of these survival curves and to determine the *p*-values. Furthermore, the maximal lifespans for all the strains were analysed by OASIS2 [42] using the modified version of the Mann–Whitney U Test, which determines the differences in the distribution tails of survival data, which affects the lifespan. This test also determines the differences in the proportion of the longevity outliers [42,43,44]. Mortality curves were plotted using the commonly used Gompertz equation [45].

Equation (4): Gompertz equation
(4)$$h(t)={Ae}^{Gt}$$

The initial mortality rate is *A*, and *G* is the exponential mortality rate coefficient. The best-fit values were determined by performing a maximum-likelihood ratio test using the flexsurv package in R [45,46]. Data on lifespan experiments for all biological replicates are summarised in Appendix A.

### 2.8. Brood Size Assay

Age-synchronous L4 staged worms (day 2 post hatch) were picked individually to seeded NGM plates and incubated at 20 °C for 24 h. About 5–7 worms per strain were used for each experiment, and the worms were transferred to fresh plates every 24 h until the cessation of egg-laying. Any worms that crawled off the plate or showed internal hatching were removed from the experiment. The plates containing the eggs were incubated at 20 °C for 3 days and then the progeny were counted. The experiment was repeated at least twice for each strain.

### 2.9. Egg Retention Assay

The egg-in-worm assay was performed according to the protocol described in [47], with a few modifications. Briefly, 15–20 L4-stage worms per strain were picked to seeded NGM plates and incubated at 20 °C for 40 h. One the day of the assay, 10 drops of bleach (2:5 *v*/*v* 4.2% NaClO:1N NaOH) were dropped on different locations of a Petri dish lid. Individual worms were transferred into each bleach drop and allowed to disintegrate for about 10 min. This gives enough time for the worm cuticle to dissolve and the eggs to remain. The eggs in utero were counted using a dissecting microscope.

### 2.10. Chemotaxis Assay

Age-synchronous worm populations were obtained by treating gravid hermaphrodites with alkaline hypochlorite (2:5 *v*/*v* 4.2% NaClO:1N NaOH) for 3 min. The reaction was stopped by adding 10 mL of M9 buffer to the sample, followed by washes to remove the residual bleach from the sample. The samples were then placed on a shaker at room temperature for 20 h. The newly hatched L1 larvae in M9 buffer were counted and about 500 L1s were transferred to seeded 9 cm NGM plates. This time point was considered to be day 1. Once the worms reached the L4 stage, the plates were washed, and the worms were transferred to FUdR-containing (25 μM) NGM plates seeded with 10% (*w*/*v*) *E. coli* OP50.

Chemotaxis was performed according to a standard protocol [48] with a few modifications. Adult worms on day 4 and day 8 were used for chemotaxis. The worms were washed off the plates with M9 buffer and transferred to a 15 mL tube. The worms were then allowed to settle, and the supernatant was removed. The washing step was repeated a total of three times to remove all residual bacteria. The worms were not centrifuged, as this could affect chemotaxis. The volatile odorant diacetyl, diluted 1:1000 (1 μL in 999 μL ethanol (100%)), was used as the test odorant for the assay. To set up a chemotaxis assay plate, the volatile odorant (1 μL) was spotted at one end of the plate, about 3 cm from the centre of the plate. The vehicle control (1 μL) (100% ethanol) was spotted on the opposite end of the plate, 3 cm from the centre, similar to the odorant spot. In addition, 5% sodium azide (1 μL) was spotted at the odorant and the control spot to immobilise the worms 15 min prior to the assay. To ensure that the worm motility towards the odorant spot was not an artefact, two plates were set up for each assay by spotting the odorant on the right side of one plate and on the left side of the other plate. The worms in buffer solution were transferred to a small piece of Whatman filter paper (grade 1), which was then inverted onto the assay plate to spot the worms at the centre of the plate. After the worms were transferred onto the agar, the Whatman paper was removed using forceps, taking care to not damage the agar surface. The plates were incubated at 20 °C for 1 h, after which they were scored, and the chemotaxis index was calculated as follows:

Equation (5): Equation for calculating chemotaxis index (CI)
(5)$$Chemotaxis\;index\left(CI\right)=\frac{\#worms\;\left(test\right)-\#worms\;(control)}{Total\;\#\;worms-\#worms\;(origin)}$$

The chemotaxis index (CI) was calculated using Excel 2010 (Microsoft, Inc., Redmond, WA, USA), and the bar graph for the chemotaxis index was plotted in GraphPad Prism 8 software (GraphPad Software, Inc., La Jolla, CA, USA) to allow for comparison between the different *C. elegans* strains.

### 2.11. Odorant Preference Associative Learning Assay

The odorant preference learning assay was performed according to the protocol described in [49], with a few modifications. Briefly, age-synchronous worm populations were obtained by bleaching gravid adults. Day 4 adult worms were washed three times with M9 buffer and transferred to an unseeded NGM plate. The worms were conditioned for 2 h by placing 2 µL of odorant (diacetyl) on the lid of the plates. In addition, another group of worms were simultaneously starved in the absence of diacetyl. Moreover, a naïve group of worms was incubated on an NGM plate in the presence of food without any odorant. At the end of the starvation period, chemotaxis assays towards diacetyl were performed on the both the naïve and the conditioned groups as described above in Section 2.9.

### 2.12. Motility Assays

Worm locomotion was measured on solid and in liquid media. To measure motility on solid media, age-synchronous worms were washed thrice with M9 buffer and placed on unseeded NGM plates (6 cm). The excess buffer was wicked away using a Kimwipe. The worms were allowed to acclimatise on the plates for 5 min, after which worm motility was recorded using an iPhone 7 Plus (Apple Inc., Cupertino, CA, USA) through Labcam from iDu Optics (New York, NY, USA), which is a microscope adaptor for the iPhone. The videos of worm motility on solid media were recorded at 25 FPS and scale of 4.39 μm/pixel, whereas the motility in liquid media was recorded at 25 FPS, 8.8 μm/pixel. Motility in liquid media was measured according to the protocol described in [46], with a few modifications. Briefly, about 10–15 worms were washed at least three times with M9 buffer and transferred to 24-well plates containing 1 mL M9 buffer in each well. The worms were allowed to acclimatise in the liquid for 1 min, after which the thrashing was recorded. Adult worms on day 4 (young), day 8 (middle-aged), and day 12 (old worms) were used for all the motility assays. Video recordings were of approximately 45 s^−1^ min duration. All the videos were then transferred to the computer, converted to .avi format, and analysed using the WormLab software system version 4.0 (MBF Bioscience, Williston, VT USA). The default track settings were used to analyse the videos, with a few exceptions. The swimming or crawling option was selected depending on the type of video being analysed, and the threshold was adjusted manually for every video.

### 2.13. Basal Slowing Response Locomotion Assays

The basal and enhanced slowing response assays were performed according to an established protocol, with a few modifications [50]. Each 6 cm NGM assay plate consisted of a ring-shaped bacterial lawn of *E. coli* strain HB101 with an inner diameter of 1 cm and outer diameter of 3.5 cm. The basal slowing response (BSR) assay was performed on well-fed worms, and the enhanced slowing response (ESR) was measured on starved worms. For BSR, well-fed worms were washed off seeded NGM plates, and half of the worms in the buffer solution were transferred to an unseeded NGM plate. The remaining half were placed at the centre of the bacterial ring in the assay plate. All the excess buffer was removed using a Kimwipe, after which the worms were allowed to acclimatise for 5 min. The worm movement on unseeded and seeded plates was recorded. The videos were recorded for 45 s^−1^ min. To measure the enhanced slowing response assay, worms were washed off seeded NGM plates and transferred to an unseeded NGM plate. The worms were then allowed to starve for 30 min to 2 h, after which some of the worms from the starved plate were picked to the centre of the bacterial ring on an assay plate and allowed to acclimatise for 5 min. The worm movement of starved worms on food and off food was recorded for 45 s to 1 min. All videos were analysed using the WormLab software. The slowing responses were measured on day 4 (young), day 8 (middle-aged), and day 12 (old) adult worms.

### 2.14. Statistical Analysis

All data are reported as mean ± standard error of the mean (SEM). For the analysis of age-associated data, we employed two-way ANOVA, with both age and genotype as variables, followed by post-hoc pairwise comparison tests. This approach allowed us to account for the differences in expression levels across the different *Aβ* strains, ensuring that genotype, as a variable, captured the variations in expression levels. For comparisons between transgenic and control strains with varied expression levels, unpaired Student’s t-tests were used to identify significant differences. All analyses were conducted using GraphPad Prism 8 software. A *p*-value < 0.05 was considered statistically significant.

## 3. Results

### 3.1. Pan Neuronal Aβ1-42-Expressing Strains Show Variation in Copy Number and Expression

Transgene copy number, which is an important determinant of expression level, was measured by qPCR. After normalisation to a single-copy reference gene, *Y45F10D.4*, the average relative *Aβ1-42* copy number in WG643 (*snb-1p::Aβ*) was 85.67 ± 3.65 copies per haploid genome, the strain WG663 (*rgef-1p::Aβ*) contained 75.86 ± 2.58 copies per haploid genome, and GRU102 (*unc-119p::Aβ*) contained 3.59 ± 0.68 copies per haploid genome (Figure 1A). As expected, the wild-type N2 Bristol, mCherry (WG731), and YFP (GRU101) control strains did not show any copies of the *Aβ1-42* transgene.

The expression levels of the *Aβ1-42* transgene were measured in these transgenic strains using an RT-qPCR assay and RNA isolated from mixed life-stage cultures (Figure 1B). After normalisation to reference genes *Y45F10D.4* and *cdc-42*, the strain WG643 (*snb-1p::Aβ*) (high copy number; *snb-1* promoter) showed the highest levels of *Aβ1-42* expression in mixed life stages (13.47 ± 0.28), followed by WG663 (*rgef-1p::Aβ*) (11.67 ± 0.26; high copy number, *rgef-1* promoter) and GRU102 (*unc-119p::Aβ*) (5.197 ± 0.91; low copy number, *unc-119* promoter). In addition, the relative log-fold expression levels of the human *Aβ1-42* transgene in the *rgef-1p::Aβ* and *snb-1p::Aβ* expressing strains were measured on day 4 (young), day 8 (middle-aged), and day 12 (old) adults (Appendix A). There was a significant difference in the *Aβ1-42* expression level between the *rgef-1p::Aβ* and *snb-1p::Aβ* strains in young adults on day 4 (*snb-1p::Aβ* >> *rgef-1p::Aβ*), but this difference disappeared in older worms.

We hypothesised that the expression levels of *Aβ1-42* in these transgenic strains would depend on a combination of the number of copies of the *Aβ* transgene integrated into the *C. elegans* genome and the intrinsic activity transgene promoter. A commonly used measure of promoter activity is fragments per transcript kilobase per million mapped reads from RNAseq data (FKPM). We used comparisons of FKPM values for the native gene in wild-type *C. elegans* from publicly available RNAseq data as an estimate of the intrinsic strengths of the promoters we used to drive transgenes in the strains described here. The FKPM for the *snb-1* gene is 192.8, for *rgef-1* it is 4.5, and for *unc-119* 22.9 (noting that *unc-119* expression is not strictly limited to the nervous system) (Appendix A). When considering expression from transgenes, the copy number of the transgene must also be taken into account. This is particularly true for the transgenes described here, all of which we generated by microinjection, which resulted in the formation of extrachromosomal arrays composed of variable numbers of the injected transgenes. Consequently, we calculated an “index of expected relative expression” as the arithmetic product of FKPM × copy number. The observed rank order of expression in either mixed life-stage cultures or in young adults (*snb-1p::Aβ* > *rgef-1p::Aβ* > *unc-119p::Aβ*) was the same as the rank order of expression predicted by the proposed index of expected expression, which combined the intrinsic promoter strength with the transgene copy number (Appendix A).

Additional transgenic *C. elegans* strains were generated to observe the pattern of transgene expression by expressing GFP driven by the same promoter fragment *snb-1* and *rgef-1* used for the *Aβ* constructs. We observed prominent pan-neuronal expression of GFP in both of our imaging experiments, consistent with previous reports characterising *snb-1* and *rgef-1* expression patterns (Appendix A) [30,31]. The GFP signal appeared diffused in the *snb-1p::GFP* construct because it did not include a nuclear localisation signal. In pan-neuronal GFP expression driven by *rgef-1,* GFP expression was evident in the head and nerve ring (H & NR), and in both nerve cords and tail neurons (T) (Appendix A).

To summarise, we obtained a panel of pan-neuronal *Aβ1-42*-expressing strains showing variation in the *Aβ* expression levels due to a combination of different pan-neuronal promoter activity and variation in the number of integrated *Aβ1-42* transgene copies. For easier presentation of the results, we labelled the strains using their promoter and Aβ status: the standard Bristol N2 strain as the wild-type N2 strain, WG643 as *snb-1p::Aβ*, the WG663 strain as *rgef-1p::Aβ*, the transgenic control strain WG731 as the pharyngeal-expressed *mCherry* control, the GRU101 control strain as the *YFP* control, and the GRU102 strain as *unc-119p::Aβ*.

Since strain WG643, or *snb-1p::Aβ*, showed the highest levels of *Aβ1-42* expression among the pan-neuronal strains, this strain was expected to show a more severe behavioural defect in comparison to the other strains if *Aβ1-42* expression level is the primary determinant of neuronal deficits, particularly in young adults when the expression level differences between the transgenes are strongest. Similarly, the GRU102 strain showed the lowest levels of *Aβ1-42* expression, and therefore, it was expected to display more moderate behavioural defects.

### 3.2. Pan-Neuronal Aβ1-42-Expressing Transgenic C. elegans Strains Show Variation in Lifespan Reduction

Measurement of lifespan is an initial step to assess whether variation in the expression of the *Aβ1-42* transgene in these strains correlates with the severity of the disease phenotype. The transgenic strains *snb-1p::Aβ* and *rgef-1p::Aβ* showed a significant reduction in lifespan in comparison to the *mCherry* control strain (*p* < 0.0001). The *snb-1p::Aβ* and *rgef-1p::Aβ* strains showed a median lifespan of 13.17 ± 0.17 and 14 ± 0.58 (S.E.), respectively, compared to the *mCherry* control WG731 (17.75 ± 0.25) (Table 1). This is a reduction in median lifespan of 26% for *snb-1p::Aβ* and 21% for *rgef-1p::Aβ*. The transgenic strain *unc-119p::Aβ* also showed a significant reduction in overall lifespan in comparison to the *YFP* control strain (*p* < 0.0001). However, the transgenic strain *unc-119p::Aβ* only showed a slight reduction in median lifespan compared to the relevant *myo-2p::YFP* (GRU101) control strain (n.s.). The median lifespan of *unc-119p::Aβ* was 15 days compared to 16 days for the *YFP* control strain, which is a ~6.25% reduction in median lifespan (Table 1). The differences in median and maximum lifespan of all *Aβ* expressing strains in comparison to their transgenic controls are listed in Table 1 and survival curves depicted in Figure 2.

The survival data were used to evaluate the differences in the rate of aging between these strains by calculating the initial mortality rate (A) and the Gompertz rate coefficient (G) from the Gompertz equation using maximum likelihood estimation. A lower G value indicates a lower rate of aging. As can be seen in Table 2, there was a significant increase in the Gompertz rate coefficient (G) of the *Aβ*-expressing strains *snb-1p::Aβ* and *rgef-1p::Aβ* in comparison to the *mCherry* control strain, indicating that the rate of aging in the *Aβ*-expressing strains was higher than in the control strain. The mortality rate doubling time (MRDT) shows that the chance of worms dying after sexual maturity doubled every 2.16 days for *snb-1p::Aβ* and every 1.79 days for the *rgef-1p::Aβ* strain, i.e., *rgef-1p::Aβ* worms were more likely to die than *snb-1p::Aβ* worms. In contrast, there was no significant difference in the Gompertz rate coefficient or in the rate of aging between the *YFP* control strain and *unc-119p::Aβ*.

### 3.3. Pan-Neuronal Aβ1-42-Expressing Strains Show Defects in Healthspan

Lifespan is a single parameter that measures the amount of time an organism survives. It does not indicate the health (physiological and functional status) of the animal. Health span can be defined as the time that an individual is active, productive, and free from age-associated diseases [51]. Measuring health span, in addition to lifespan, is feasible in *C. elegans*. *C. elegans* exhibits several behaviours that are important for its survival and reproduction and are therefore potentially useful indicators of health span because those behaviours are indicative of functionality: locomotion (crawling and swimming), feeding (pharyngeal pumping), defecation, egg-laying, mating, and its ability to sense and respond to chemical, mechanical, and thermal stimuli. Therefore, health span is a concept that attempts to capture the overall state of physical health by measuring complex phenotypes such as reproductive output and coordinated movement. Two such phenotypes in *C. elegans* are egg-laying rate (eggs laid per unit time) and maximum crawling or swimming speed (on solid or in liquid media, respectively). We sought to test whether variation in the expression of *Aβ1-42* influenced the severity of health span indicators such as fecundity and maximum speed.

Reproduction is an excellent indicator of an organism’s general physiological health.

Physiological decline begins earlier when reproduction is adversely affected while the animal is relatively young. In contrast, lifespan assays measure the duration of survival, which reflects the overall health and physiological state of the organism after the reproduction period has ended. Reproductive aging in *C. elegans* hermaphrodites is characterised by a progressive age-related decline in physiological function beginning on the 5th day and ceasing on the 10th–14th day of adulthood [52,53,54]. Egg-laying is a well-studied aspect in *C. elegans* physiology and an important reproductive indicator of health span [55]. We sought to measure the egg-laying rate in these transgenic strains to determine the effect of variation in *Aβ1-42* expression on reproduction. First, the total number of progeny produced per day during the reproductive span of the worm was counted (Figure 3A) and the total brood size was estimated from these data (Figure 3B). The *snb-1p::Aβ* strain showed a significant reduction in the number of progeny produced on day 6 (*p* = 0.0001) and day 7 (*p* = 0.0034), thereby leading to a significant reduction in the total brood size (225 ± 8.053) in comparison to the *mCherry* control strain (286 ± 10.37). Although the *rgef-1p::Aβ* strain showed a significant reduction in the number of progeny produced on day 6 (*p* = 0.0119) and day 7 (*p* = 0.0251), there was only a moderate reduction in the total brood size of *rgef-1p::Aβ* (265 ± 12.44) in comparison to the *mCherry* control strain WG731. This difference in the total brood size between the *rgef-1p::Aβ* and *mCherry* control strain was not statistically significant (*p* = 0.76). Similarly, the strain *unc-119p::Aβ* showed a reduction in brood size (248.8 ± 14.72) compared to the *YFP* control strain (282.8 ± 11.79); however, the difference in the total brood size between the *unc-119p::Aβ* and the *YFP* control strain was not statistically significant (*p* = 0.87).

Secondly, the number of eggs retained in utero was measured (Figure 3C). Egg laying has a neurological basis, and thus, discriminating between differences in egg production and egg laying is important for assessing whether observed strain variation in brood size is because of changes in the egg-laying neurological circuit [56]. Egg retention reflects a balance between how many eggs are produced and how many are actually laid [47]. The number of eggs retained was significantly reduced in *snb-1p::Aβ* (*p* = 0.0192) and *rgef-1p::Aβ* (*p* = 0.0419) in comparison to the *mCherry* control strain, whereas no difference was seen in the *Aβ1-42*-expressing strain *unc-119p::Aβ* (*p* = 0.447) compared to the *YFP* control strain. Therefore, the ability to lay eggs was not affected but the rate of egg production was reduced, which resulted in an overall reduction in brood size in all the *Aβ*-expressing transgenic strains in comparison to the transgenic controls.

Additionally, maximum speed on solid media is considered to be an indicator of health span and was measured in these strains [57]. The *snb-1p::Aβ* did show a significant reduction in maximum speed on day 12 (*p* = 0.039) by the non-parametric Mann–Whitney test. In contrast, the *rgef-1p::Aβ* strain did not move faster and the *unc-119p::Aβ* strain did not show changes in maximum speed relative to the relevant control strain (Figure 3D). Therefore, the higher *Aβ*-expressing strain *snb-1p::Aβ* showed a more severe phenotype in comparison to other *Aβ*-expressing strains for both health span phenotypes that were assessed.

### 3.4. Pan-Neuronal Aβ1-42-Expressing Strains Show a Significant Reduction in Motility Parameters on Solid and Liquid Media

Previous studies have shown that movement on solid and in liquid media measures different aspects of health and behaviour since the kinematics of swimming is distinct from that of crawling [58]. Movement on solid media (crawling) is characterised by a sinusoidal posture, whereas in liquid media, worms show a C-shaped posture [59,60,61]. Several motility parameters in addition to maximum speed were measured in the *Aβ*-expressing transgenic strains on solid and in liquid media to determine whether differences in the levels of *Aβ* expression result in variation in specific components of worm motility. Firstly, mean speed on solid media was assessed. There was an age-related decline in the mean speed in all the transgenic strains on solid media (Figure 4A). The *snb-1p::Aβ* and *rgef-1p::Aβ* showed a significant reduction in mean speed even in young adult animals on day 4, in addition to day 8 and day 12. Therefore, the moderate and high *Aβ*-expressing strains showed age-related motility defects on solid media in comparison to the transgenic *mCherry* control. On the other hand, the low *Aβ*-expressing strain (*unc-119p::Aβ*) did not show significant changes in mean speed on solid media in comparison to the YFP transgenic control.

Furthermore, motility in liquid was assessed in these strains. The mean swimming speed was significantly lower on day 4 (*p* = 0.0348), day 8 (*p* = 0.0001), and day 12 (*p* < 0.0001) in the transgenic *Aβ1-42* expressing strain *snb-1p::Aβ* compared to the *mCherry* control strain (Figure 4B). Similarly, the *rgef-1p::Aβ* strain showed a significant age-related decline in mean swimming speed on day 4 (*p* = 0.0032), day 8 (*p* = 0.0025), and day 12 (*p* = 0.0006) relative to the *mCherry* control strain. On the other hand, the *unc-119p::Aβ* strain showed a significant decline in the mean swimming speed only on day 4 (*p* = 0.0028) in young adults compared to the *YFP* control strain. The wave initiation rate, which is defined as the number of body waves initiated from the head or tail per minute, was also measured (Figure 4C). There was a rapid decline in the wave initiation rate of the *rgef-1p::Aβ* on day 4 (*p* < 0.0001), day 8 (*p* = 0.0051), and day 12 (*p* = 0.026) in comparison to the *mCherry* control strain. However, the *snb-1p::Aβ* strain showed a rapid decline in the wave initiation rate on day 8 (*p* < 0.0001) and day 12 (*p* = 0.0010). In contrast, there was no difference in the wave initiation rate between the *YFP* control strain and *unc-119p::Aβ*. The activity index, which is a measure of how vigorously the worm bends, was also analysed. There was an age-related decline in the activity of all the *Aβ1-42* expressing strains (Figure 4D). The *rgef-1p::Aβ* showed a significant decline in activity starting on day 4 (*p* = 0.0018) in comparison to the *snb-1p::Aβ*, which showed a reduction in activity on day 8 (*p* = 0.0005) relative to the *mCherry* control. In addition, there was a decline in the activity of the *unc-119p::Aβ* worms on day 4 (*p* = 0.037), but not on day 8 (*p* = 0.35) or day 12 (*p* > 0.9999). Curvature-based motility analysis using parameters such as brush stroke, dynamic amplitude, and curling of the worms was also performed (Appendix A). The dynamic amplitude gives a sense of whether the body bends are deep or flat and how much stretching effort occurs in a single body bend (Appendix A). Additionally, the curling parameter determines the percentage of time the animal spends in the bent state and can detect changes in the curl pattern of swimming worms (Appendix A). The low *Aβ*-expressing strain *unc-119p::Aβ* showed a significant reduction in curling on day 12 (*p* < 0.0001) compared to the *YFP* control strain (Appendix A), which indicates that these worms were experiencing issues with movement control, possibly due to neuromuscular impairment as they aged. Additionally, the brush strokes refer to the sweeping movements made by the worms as they navigate their environment and gives an indication of the depth of the worm movement and extent to which the worms have stretched in a given body bend (Appendix A). There, *snb-1p::Aβ* in comparison to the transgenic control strain showed a significant reduction in brush strokes on day 12 (Appendix A). A reduction in brush strokes could indicate a decline in exploratory behaviour or underlying issues such as neuromuscular deficits.

The low *Aβ*-expressing strain *unc-119p::Aβ* showed a significant decline in swimming parameters, mean swimming speed, and activity compared to the *YFP* control strain on day 4 (Figure 4B,D) and in the curling parameter in older worms on day 12 (Appendix A). As the decline in motility parameters was more pronounced (relative to the transgenic control) in high and moderate *Aβ1-42*-expressing strains *snb-1p::Aβ* and *rgef-1p::Aβ* but not in *unc-119p::Aβ* for most of the motility parameters tested, the motility defect was severe in high and moderate *Aβ*-expressing strains. In summary, all of the *Aβ*-expressing strains showed an age-related decline in motility parameters in liquid media, but the severity of the motility deficit correlated with the level of *Aβ* expression.

### 3.5. Pan-Neuronal Aβ1-42-Expressing Strains Show Defects in Neuronal Function

Another behaviour used to assess age-associated neuronal dysfunction is olfaction. Since olfaction in worms is mediated by a specific subset of chemosensory neurons, a well-characterised volatile odorant, diacetyl, was used for chemotaxis assays [48]. This assay was used to measure any differences in the chemotactic responses between these strains as a result of variation in the levels of neuronal *Aβ1-42* expression. The chemotaxis index of the worms towards diacetyl was measured in young (day 4) and middle-aged (day 8) adults. In addition, worm mobility was measured on the chemotaxis plates by calculating the percentage of worms that moved away from the origin after 1 h. A schematic illustration of a chemotaxis assay is shown in Figure 5A. Chemotaxis assays measure the fraction of the worms that are attracted to a particular chemical in a given time interval. This fraction is represented as a chemotaxis index, and the values obtained range from +1 to −1, wherein +1 indicates maximum attraction and −1 indicates maximum repulsion.

There was a significant decrease in the chemotaxis index of the *snb-1p::Aβ* strain towards diacetyl on day 4 (CI 0.58 ± 0.10) in comparison to the transgenic control strain (*p* < 0.01, unpaired t-test) (Figure 5B). Although the *snb-1p::Aβ* strain also showed a reduction in the chemotaxis index towards diacetyl on day 8 (CI 0.41 ± 0.13) compared to the transgenic control (CI 0.65 ± 0.063), the difference was not statistically significant. In comparison, the transgenic strain *rgef-1p::Aβ* did show reduced chemotactic ability on day 4 (CI 0.66 ± 0.085) and day 8 (CI 0.37 ± 0.10), although the data were not statistically significant compared to those of the *mCherry* control strain. Furthermore, the *Aβ1-42*-expressing strain *unc-119p::Aβ* also showed a significant reduction in the chemotactic ability towards diacetyl in middle-aged worms on day 8 (CI 0.50 ± 0.082) compared to the transgenic *YFP* control (CI 0.77 ± 0.07) (*p* = 0.045, unpaired t-test), but not on day 4 (*p* = 0.36). There was an age-related decline in the mobility of all the worm strains on day 8, but there was no significant difference between the mobility of the transgenic *Aβ1-42*-expressing strains in comparison to the controls (Figure 5C). Therefore, the differences in the chemotaxis obtained between the strains were due to chemotactic defects and not due to movement per se.

In summary, the worms moved away from the origin towards the odorant, but in general the chemotaxis indices were not statistically different than those of the relevant controls. The exceptions that were statistically significant were *snb-1p::Aβ* on day 4 and *unc-119p::Aβ* on day 8. It is important to note that these mobility data only indicate the ability of the worms to move towards the odorant in a given time period, and do not detect any motility defects between worm strains.

In order to study the effects of *Aβ* expression on the learning and memory of the worms, food-odorant association-based learning assays were conducted using the high *Aβ1-42*-expressing strain *snb-1p::Aβ*. In this assay, young adult worms on day 4 were starved in the presence of the attractant diacetyl for 2 h, and standard chemotaxis assays towards diacetyl were performed after the incubation period. The groups starved in the absence and presence of diacetyl were denoted as diacetyl “−“ and diacetyl “+” in Figure 5D. The naïve group consisted of well-fed worms incubated in the absence of the odorant. The N2 strain and the transgenic *mCherry* control strain WG731 showed an aversive response by a significant reduction in chemotaxis towards diacetyl after the starvation period. Although the transgenic *Aβ*-expressing strain group starved in the presence of diacetyl showed a moderate reduction in chemotaxis towards diacetyl after the starvation, the difference between the groups that were starved in the presence and absence of the odorant was not significant.

Moreover, to further validate the severity of the phenotypes observed in our human *Aβ*-expressing strains, we generated a non-toxic mouse *Aβ1-42* transgene-expressing strain driven by same pan-neuronal promoter *snb-1*. The mouse *Aβ1-42* peptide, while capable of aggregation, is significantly less prone to forming the toxic oligomers characteristic of the human *Aβ1-42* peptide. The mouse *Aβ*-expressing strain does not exhibit severe behavioural defects in lifespan, brood size, or motility compared to the human *Aβ1-42*-expressing strain or the transgenic *mCherry* control strain (Appendix A). This suggests that the toxic phenotypes we observed are indeed specific to the human *Aβ1-42* peptide, rather than a general consequence of expressing any aggregating protein or the result of promoter competition.

### 3.6. Pan-Neuronal Aβ1-42-Expressing Strains Exhibit Behavioural Deficits Potentially Linked to Dopaminergic Signalling

Experienced-based locomotion assays are used to study deficits in neurotransmission. The behaviour of worms has been shown previously to change in the presence of food in the environment. For instance, well-fed worms slow down when reintroduced to a bacterial lawn (seeded plate). This slowing response, also known as the basal slowing response (BSR), is experience-based and shown to be dependent on the dopaminergic neuronal pathway [50,62]. The diminished slowing response of well-fed worms on food suggests that there may be impairment in dopaminergic signalling. To determine deficits in dopaminergic signalling, the high and moderate *Aβ*-expressing strains *snb-1p::Aβ* and *rgef-1p::Aβ* were used for the basal slowing response assays. In young adults on day 4, all the transgenic *C. elegans* strains showed a significant reduction in the number of body bends in the presence of food (seeded plate), except the control strain CB1112 *cat-2* (Figure 6A). The strain *cat-2* is a negative control strain that carries a loss of function mutation in the *cat-2* (tyrosine hydroxylase homolog) gene that inactivates its ability to slow down in the presence of food and is defective in dopamine synthesis.

On day 8, this slowing response diminished for both the *Aβ1-42*-expressing strains *snb-1p::Aβ* and *rgef-1p::Aβ* in comparison to the *mCherry* control strain (Figure 6B). These results indicate that there may have been alteration in dopaminergic signalling in the high and moderate *Aβ*-expressing strains *snb-1p::Aβ* and *rgef-1p::Aβ* as the worms aged.

## 4. Discussion

Although the *Aβ* peptide has been proposed to play a role in the pathogenesis of AD, direct evidence that quantitatively links *Aβ* concentration to disease severity is lacking. Low levels of *Aβ* are protective; however, high levels of *Aβ* resulting from an imbalance between production and clearance lead to an accumulation of *Aβ* and an increase in *Aβ* aggregation, which triggers a pathogenic cascade, ultimately leading to disease [17]. According to this hypothesis, there may be a correlation between *Aβ* concentration and the initiation, progression, and/or endpoint of the disease. To explore this idea, *C. elegans* strains expressing varying levels of *Aβ1-42* were generated using two different pan-neuronal promoters, *rgef-1* and *snb-1*, and were compared with a previously published strain in which the *Aβ1-42* peptide was expressed using the pan-neuronal *unc-119* promoter [27]. The *unc-119p::Aβ* strain was previously shown to have a reduced lifespan, coupled with deficits in egg-laying and locomotory behaviours. The early onset of middle-aged behaviours in these animals correlated with metabolic decline and electron transport failure that preceded *Aβ* toxicity [27]. The pan-neuronal *Aβ1-42* expressing transgenic *C. elegans* strains described in this study were used to test the hypothesis that the severity of the disease phenotype is correlated with the levels of *Aβ1-42* expression. These strains showed varying levels of the *Aβ1-42* expression and also showed variation in the severity of some, but not all, of the behavioural phenotypes that were measured. Overall, all the *Aβ*-expressing strains showed reduced longevity, impaired egg laying, and an age-related decline in motility in liquid media, accompanied by subtle defects in chemotaxis and potential impairment in dopaminergic signalling. In addition, the non-toxic mouse *Aβ*-expressing strain (*snb-1p::mouseAβ)* did not show any severe behavioural defects in comparison to the transgenic human *Aβ*-expressing strain (*snb-1p::Aβ*) (Appendix A). The Aβ peptide found in mice contains three amino acid substitutions within this metal binding domain (Arg5Gly, Tyr10Phe, His13Arg) and significantly decreases the metal binding activity and peroxide production, which suggests that the mouse Aβ is not as intrinsically pathogenic as the human *Aβ* [63]. These results support the specificity of the observed toxic effects of the human *Aβ1-42* peptide.

### 4.1. Transgenic Aβ1-42-Expressing C. elegans Strains Show Variation in Aβ Expression

The rank order of expression observed amongst the transgenic strains we describe here is the same as that predicted by the index of expected relative expression, i.e., *snb-1p::Aβ* > *rgef-1p::Aβ* > *unc-119p::Aβ*. We propose that the observed differences in transcript abundance give rise to variation in the abundance of *Aβ* peptide, particularly in younger worms. When comparing the *Aβ* expression levels by RT-qPCR, there was a 100-fold difference in *Aβ* expression between *snb-1p::Aβ* and *rgef-1p::Aβ*. In addition, *rgef-1p::Aβ* showed 200% higher *Aβ* expression than *unc-119p::Aβ*, which showed the lowest levels of *Aβ* expression (Figure 1B). That said, our index incorporates the intrinsic promoter strength based on the FPKM value, which is a direct measure of transcript abundance, while *Aβ* expression as measured by RT-qPCR is a relative measure of transcript abundance and thus dependent on the “housekeeping” transcript(s) that is/are used. In addition, direct comparisons of *snb-1p::Aβ* and *rgef-1p::Aβ* strains with the *unc-119p::Aβ* strain are complicated because the latter was constructed in a different genetic background, has a different transgenic marker that mediates the quantitative variation in *Aβ* expression, and varies in its intrinsic promoter activity. Another factor potentially influencing the *Aβ* expression in these transgenic strains is the location of integration sites.

Our categorisation of the transgenic lines as high, moderate, and low expressers is based on expression levels measured in mixed populations of worms. However, these expression levels may diverge with age, potentially influencing the phenotypic outcomes. Additionally, post-transcriptional mechanisms could contribute to the observed differences in expression, independent of promoter activity. Regardless of the quantitative differences in expression (or their cause), if the *snb-1p::Aβ* strain with the highest levels of *Aβ* expression shows a more severe behavioural deficit than the *rgef-1p::Aβ* strain while the *unc-119p::Aβ* strains shows only subtle behavioural deficits, this supports the hypothesis that levels of *Aβ* expression correlate with the severity of the disease phenotype, particularly behavioural deficits such as lifespan, motility in liquid, and fecundity.

### 4.2. Pan-Neuronal Aβ1-42-Expressing Strain Shows Premature Death and Reduction in Fecundity

All the transgenic *Aβ1-42*-expressing strains showed a significant reduction in median lifespan. The rank order of the reduction in lifespan was the same as the rank order for transcript abundance: *snb-1p::Aβ* (~ 26%) > *rgef-1p::Aβ* (~21%) > *unc-119p::Aβ* strain (~6%) (Table 1). Therefore, the reduction in median lifespan mirrored the levels of *Aβ* expression. Similarly, the rate of aging, measured by the Gompertz equation, was greater in the higher *Aβ*-expressing strains *snb-1p::Aβ* and *rgef-1p::Aβ* (Table 2) than in the relevant control strain or in the *unc-119p:: Aβ* strain. However, the *unc-119p:: Aβ* strain did not age significantly faster than its control. These results indicate that increased levels of *Aβ* in the *C. elegans* neurons cause the worms to age faster, whereas low levels of *Aβ* may influence lifespan without necessarily changing the rate of aging.

In addition, all the *Aβ*-expressing strains showed a reduction in brood size (Figure 3B). However, this reduction in brood size was only significant for the highest *Aβ*-expressing strain, *snb-1p::Aβ*. In addition, there was possibly an impairment in the rate of egg production in these strains, which would result in a reduction in the brood size. That said, reproductive span was not correlated with age-related declines in motility, pharyngeal pumping, or survival probability, which suggests that reproductive span may be regulated independently of these processes [64]. This correlation may change if the hermaphrodite worms are mated with males, which replenishes their sperm, thereby increasing production 3-4-fold and doubling the reproductive span. Thus, although fecundity is accepted as a measure of health span, it may not necessarily correlate with specific age-related neuronal deficits and does not allow for the determination of the underlying causes of change in those measures of health span. It is also important to note that all the young adult *Aβ*-expressing worms lay eggs at the rate close to the wild type, and the rate of egg production was lower than the transgenic control only on day 6 and day 7 of adulthood. Further experimentation is required to validate these results. For instance, ectopic expression of serotonin to stimulate egg laying could be used to test whether the worm’s response to the treatment restores the egg-laying deficit [65,66]. As the expression of *Aβ* in the worm nervous system has been shown to impact various physiological functions and its effect on fecundity is not yet fully understood, the observed reduction in fecundity in our *Aβ*-expressing worms may reflect broader neurodegenerative effects, but further investigation is required to thoroughly establish this relationship.

### 4.3. Pan-Neuronal Aβ1-42-Expressing Strains snb-1p::Aβ and rgef-1p::Aβ Show a Severe Decline in Motility

Motility has also been suggested as an indicator of health span [57]. There was a rapid age-related decline in several motility parameters, generally in the same rank order of severity as described above for lifespan and fecundity parameters. When comparing motility on solid media, only the high *Aβ*-expressing strains *snb-1p::Aβ* and *rgef-1p::Aβ* showed significant declines in crawling parameters such as mean speed, and only the *snb-1p::Aβ* strain showed a significant decline in maximum speed, a measure that has been correlated with health span. The low *unc-119p Aβ*-expressing strain did not show a significant difference in any motility parameter on solid media relative to the control strain. When looking at motility parameters in liquid media, the *snb-1p::Aβ* and *rgef-1p::Aβ* strains showed a decline in swimming parameters such as the mean swimming speed, activity, and wave initiation rate (Figure 4). The low *Aβ*-expressing strain *unc-119p::Aβ* also showed a decline in all of these swimming parameters, except for the wave initiation rate. Although the wave initiation rate in this strain showed a decline from day 4 to day 8, the change was similar to the one seen in the transgenic control (Figure 4C). It is easier to detect worm motility defects in liquid. Worms crawling on an agar surface encounter mechanical resistance that is 10,000-fold greater than that encountered while swimming in liquid, and this resistance may mask any subtle defects in motility. Despite the higher resistance of crawling on agar versus swimming, swimming in liquid is more energetically demanding than crawling, and there is increase in the metabolic rate of swimming worms [67]. Consistent with the defects in swimming parameters for the *unc-119p::Aβ* strain, the low *Aβ*-expressing strain *unc-119p::Aβ* has been reported to show an ATP deficit at young age [27]. We hypothesise that a more severe energy/ATP generation defect would be observed in the *snb-1p::Aβ* and *rgef-1p::Aβ* strains. *S*ince a decline in motility is an early effect of aging, it may eventually compromise other behavioural responses [68].

Locomotion is an important behavioural output in *C. elegans* regulated by a neural circuit generating sinusoidal bends in response to diverse sensory cues. The sensory cues are integrated by the sensory neurons, which then transmit this information to the interneurons that mediate the flow of this information to the motor neurons and finally to the muscle [20]. Given this context, behaviour is initiated at the neurons upstream of the pathway and executed in the muscle, and therefore, locomotory deficits may arise because of defects in muscle or neurons. For instance, the “uncoordinated” phenotype is an abnormal locomotion phenotype that causes twitching, halting, odd jerky motions, curling or kinking motion, and partial or rigid paralysis, and may arise as a result of mutations in genes expressed in muscles, neurons, or other tissues. There are 111 Unc genes in *C. elegans*, but 71 of those primarily affect the nervous system and not the muscle [69]. The locomotory deficits observed in the reported strains are neuronal deficits resulting from *Aβ* expression in the neurons upstream of this pathway. To test this more explicitly, *Aβ* expression may be targeted to specific neuronal subsets to assess the effects on the severity of the different phenotypes.

### 4.4. Defects in Chemotaxis and Learning Behaviours Observed in All Aβ1-42-Expressing Strains

Olfaction is affected in AD and is considered to be an early indicator of disease in humans [70]. All the pan-neuronal *Aβ1-42* expressing strains showed subtle defects in chemotaxis towards diacetyl. The *snb-1p::Aβ* and *rgef-1p::Aβ* strains showed reduced chemotaxis towards diacetyl on day 4, whereas the decline in chemotactic activity was delayed until day 8 in the *unc-119p::Aβ* strain, demonstrating again that phenotypic severity and/or age of onset are correlated with *Aβ* expression level. A plausible explanation for this observation is that there is a threshold of *Aβ* expression at which the phenotype is detectable, and *snb-1p::Aβ* reaches this threshold for chemotaxis defect earlier because it accumulates *Aβ* faster. Thus, changes in olfaction are early indicators of age-related cognitive decline due to *Aβ* in worms, as in humans, because human olfaction is sensitive to the presence of low concentrations of *Aβ* [70].

High-level cognitive abilities across *C. elegans* strains are lost much earlier than basic motility and chemotaxis abilities. Therefore, deficiencies in the worms’ learning behaviour can be detected early on [71,72,73,74]. All the memory assays were performed on young adults (day 4) since the chemoreception abilities should be intact on day 4 before the loss of chemotaxis. Food–odour associative learning is a food-dependent behaviour mediated by serotonin, which informs the food status of the environment to the nervous system [75]. The first associative-learning assay used adaptation suppression, as starving wild-type worms in the presence of diacetyl reduces the chemotaxis of the worms towards the odorant. Studies have shown that high and low concentrations of diacetyl induce distinct learning mechanisms [76]. The *Aβ*-expressing worms showed reduced adaptation suppression when starved in the presence of diacetyl and therefore showed defects in associative learning (Figure 5D). Another study performed the same assay using benzaldehyde and found this associative learning behaviour to be mediated by serotonergic signalling [77]. A similar learning defect was reported for the *Aβ*3-42-expressing strain when using diacetyl in the learning assay [49].

The nervous system allows the animal to respond flexibly to changes in the environment, which involves alteration of the properties of neurons and synapses caused by the action of neuromodulators such as dopamine and serotonin [50]. Several studies have clearly proven the critical role played by dopamine as a neuromodulator in altering the intrinsic properties within circuits (both pre- and post-synaptically), thereby regulating behaviours such as cognition, egg laying, locomotor activity, and learning [78,79,80,81,82]. The basal slowing response in *C. elegans* is mediated by dopaminergic neurons that transduce the mechanosensory stimulus (caused by bacteria and surface tension or pressure exerted by the agar surface on the cuticle) through interneurons and motor neurons to the motor circuit, thereby affecting locomotion [50]. In *C. elegans,* the basal slowing response was sensitive to even small amounts of *Aβ*: All the *Aβ*-expressing strains showed a diminished basal slowing response on day 8 (Figure 6B). These findings support the idea that *Aβ* may contribute to synaptic dysfunction, potentially impacting neurotransmitter deficit signalling. However, the deficits in dopaminergic signalling should be interpreted with caution, as broader neuronal or sensory effects could also be involved (Figure 6). Synaptic dysfunction in AD is considered to be an early hallmark of the disease [49]. These motility-related behaviours are noisy, and experiments to reveal subtle defects that may be the first signs of *Aβ* toxicity will require large sample sizes. Several studies have reported that serotonergic and dopaminergic neurons are susceptible to aging in humans, specifically in age-related diseases such as AD [83]. To provide a more comprehensive assessment of potential deficits in synaptic transmission, an aldicarb assay could be performed [84]. The aldicarb assay, which evaluates the ability of the neurons to release neurotransmitters in response to a stimulus, could help elucidate whether the observed behavioural deficits are associated with synaptic transmission and could strengthen our understanding of the in vivo toxicity associated with the *Aβ* species.

Table 3 shows the ranking of the *Aβ*-expressing strains according to the severity of the disease phenotype. There were notably statistically significant differences between these strains for some, but not all, phenotypes. Nevertheless, the *Aβ*-expressing strains were ranked on the basis of phenotype severity, wherein 1 indicates a mild phenotype and 3 indicates a severe phenotype, which is in turn correlated with the level of *Aβ* transcript abundance. As shown in Figure 7, the relation between *Aβ* concentration and disease severity could be explained in two ways: (i) a continuous dose–response curve in which is there is gradual increase in disease severity as the concentration of *Aβ* increases (Figure 7A) or (ii) a threshold response in which there is a range of *Aβ* concentrations at which there is no phenotype, then a step-wise transition to disease as a threshold concentration is exceeded (Figure 7B). The *snb-1p::Aβ* and *rgef-1p::Aβ* strains we describe here had very similar transcript abundance. This means it was not possible to differentiate between a continuously variable concentration-to-phenotype relationship (Figure 7A) and the stepwise threshold (Figure 7B). Strains with intermediate levels of expression are required to investigate this further. It is also possible that different phenotypes will have different relationships between severity and concentration.

It was also observed that some phenotypes, such as deficits in neurotransmitter signalling, were more sensitive to the presence of *Aβ*. These results therefore provide insight into the relationship between *Aβ* toxicity and the severity of the behavioural deficit. Future work could include generating several transgenic *C. elegans* strains expressing levels of the *Aβ* transgene that span the range between *unc-119p:: Aβ1-42* and *rgef-1p:: Aβ1-42* to better correlate expression with the severity of the behavioural deficit. This could be achieved by using gene-editing tools such as MosSCI or CRISPR to integrate a specific number of *Aβ* transgene copies driven by these promoters at a specific location in the *C. elegans* genome [85]. Alternatively, a titration experiment could be conducted by microinjecting the same promoter construct at different concentrations to generate strains containing varying copies of the *Aβ* transgene, thereby resulting in a range of *Aβ* expression. The strains we have reported with differing *Aβ* expression levels are intended as a starting point for further investigation.

## 5. Conclusions

This is the first study to demonstrate that varying expression levels using different promoters, and a variable copy number of different integrated arrays establishes a quantitative dose response between *Aβ* expression and AD-related phenotypes. These data support the conclusion that there is a strong positive correlation between *Aβ* concentration and the initiation and progression of the disease, rather than the end point of the disease. Therefore, *C. elegans* serves as a good model to study the relationship between disease progression and *Aβ* expression because the severity of these behavioural deficits is correlated with the quantitative expression of *Aβ*. Additionally, these data provide preliminary support for the hypothesis that some behaviours are more sensitive indicators of the disease, such as the basal slowing response assay in which all the *Aβ*-expressing strains show diminished slowing response irrespective of the *Aβ* concentration.

## Figures and Tables

**Figure 1 cells-13-01598-f001:**
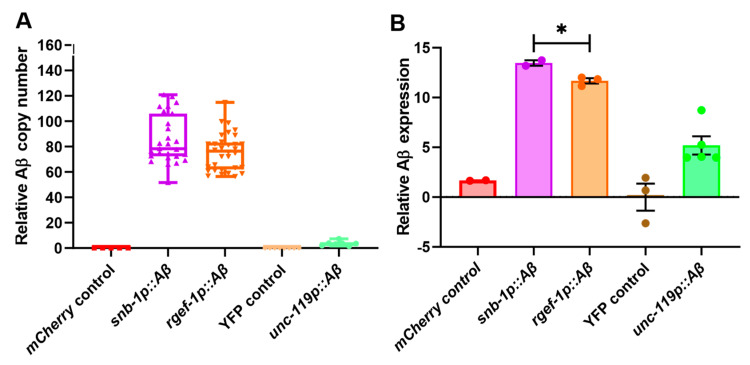
Comparison of *Aβ* copy number and expression levels in the pan-neuronal *Aβ* strains. (**A**) Copy number assay showing the number of copies of the integrated *Aβ* transgene in each strain relative to the reference gene *Y45F10D.4* (copy number 1 per haploid genome) (N = 3 replicates, n = 8 worms/replicate). (**B**) Relative log *Aβ* expression at the transcript level in the nematodes (mixed life stages), normalised to reference genes *Y45F10D.4* and *cdc-42* (N = 2–3 replicates, n = 200–300 worms/replicate). An unpaired t-test was used to test for significant differences between *Aβ*-expressing worms *snb-1p::Aβ* and *rgef-1p::Aβ*. * *p* < 0.05. All bar graph data are reported as mean ± standard error of the mean (SEM).

**Figure 2 cells-13-01598-f002:**
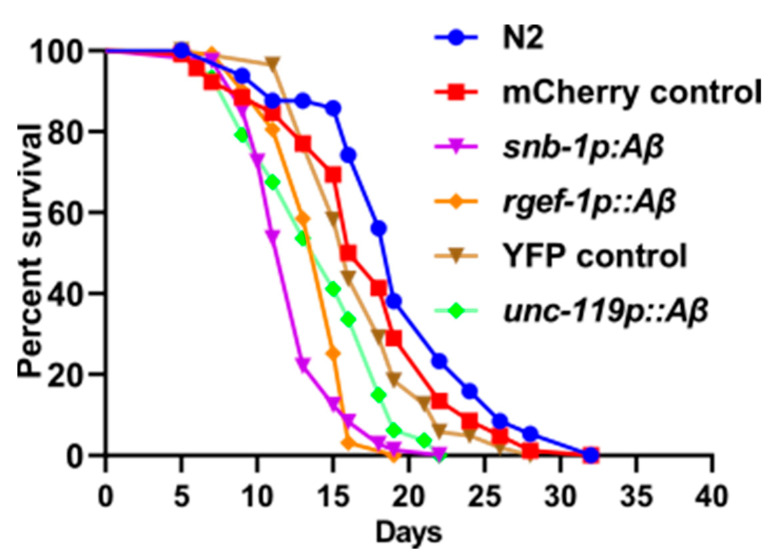
Transgenic *C. elegans* strains expressing *Aβ* peptide in neurons show reduction in life span. Representative Kaplan–Meier survival curves of one biological replicate (n = 120).

**Figure 3 cells-13-01598-f003:**
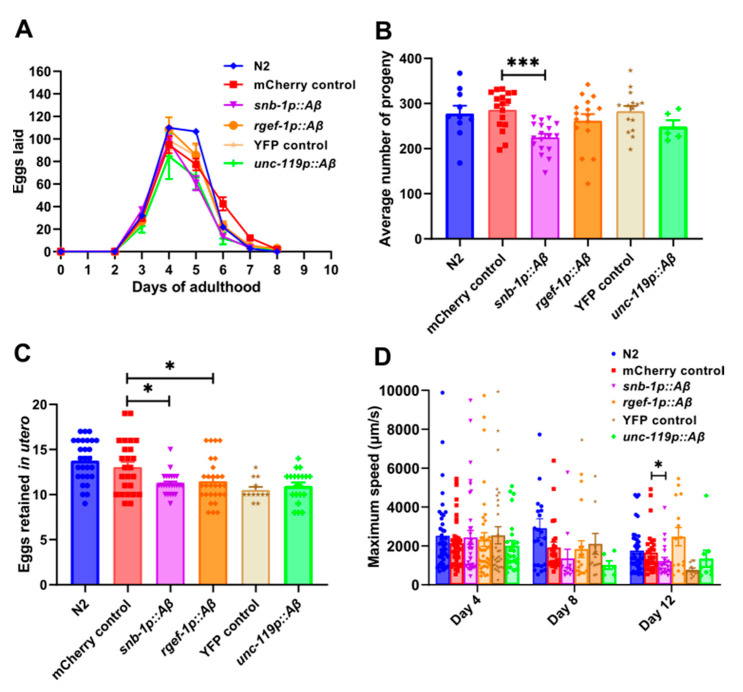
Reduction in reproductive output and maximum speed as indicators of health span in *Aβ*-expressing strains. (**A**) Number of eggs laid per worm per day during the reproductive span of the animal (n = 3, 5–7 worms/replicate). (**B**) Total brood size (n = 3, 5–7 worms/replicate). (**C**) Mean number of eggs retained per worm in utero (n = 3, 15–20 worms/replicate). (**D**) Maximum speed on solid media (n = 3, 5–15 worms/replicate). An unpaired t-test was used to test for significant differences between *Aβ*-expressing worms and transgenic controls. ns: not significant, * *p* < 0.05, *** *p* < 0.001. All bar graph data are reported as mean ± standard error of the mean (SEM).

**Figure 4 cells-13-01598-f004:**
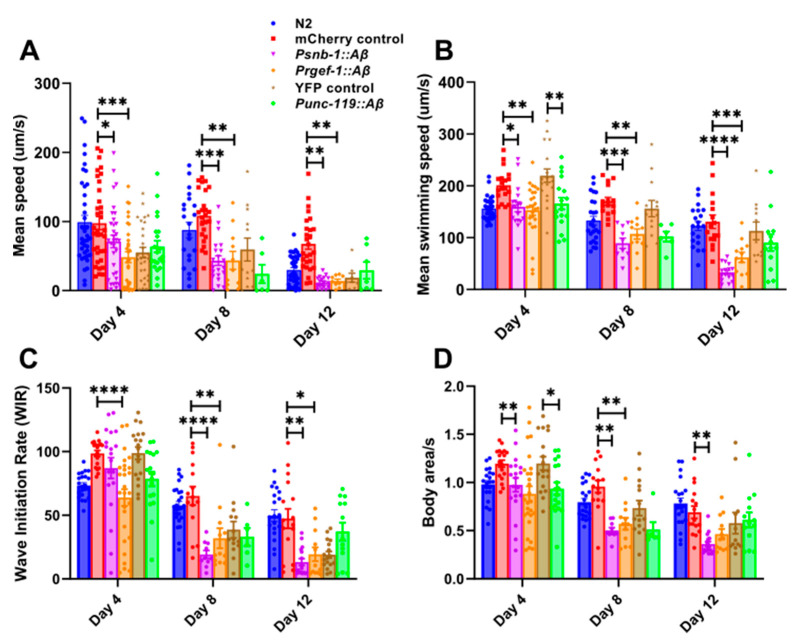
Pan-neuronal *Aβ*-expressing strains show age-dependent reduction in motility on solid and in liquid media. (**A**) Mean speed on solid media (μm/s) (n = 3, 5–15 worms/replicate). (**B**) Mean swimming speed (μm/s). (**C**) Wave initiation rate (WIR). (**D**) Activity in liquid (body area/s) (n = 2, 5–15 worms/replicate). All data analysed by two-way ANOVA followed by post hoc Tukey multiple comparisons test. The *snb-1p::Aβ* and *rgef-1p::Aβ* were compared to the transgenic *mCherry* control strain, and the *unc-119p::Aβ* was compared to the transgenic *YFP* control strain. * *p* < 0.05, ** *p* < 0.01, *** *p* < 0.001, **** *p* < 0.0001. All bar graph data are reported as mean ± standard error of the mean (SEM). The genotypes shown in the key in panel A are the same across multiple panels.

**Figure 5 cells-13-01598-f005:**
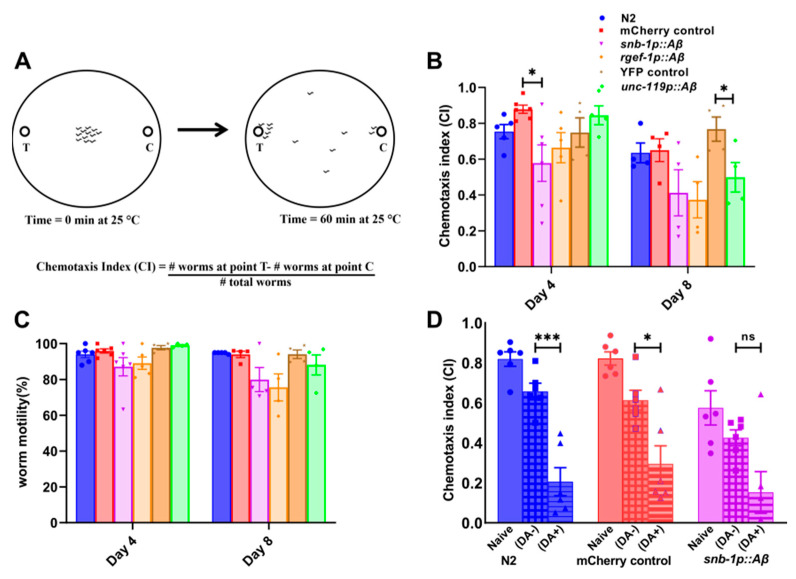
Pan-neuronal *Aβ*-expressing strain shows deficits in chemotaxis and learning. (**A**) Schematic illustration of a chemotaxis assay. (**B**) Bar graph showing variation in chemotactic response of *Aβ*-expressing transgenic *C. elegans* strains towards volatile odorant diacetyl in young (day 4) and middle-aged (day 8) worms. An unpaired t-test was used to test for significant difference in chemotactic response between transgenic control and *Aβ*-expressing strains. (**C**) Percentage motility of young (day 4) and middle-aged (day 8) worms on diacetyl chemotaxis plates. (**D**) Food-odorant association-based learning assay using volatile odorant diacetyl (n = 6, 100–200 worms per biological replicate). An unpaired t-test was used to test for significant difference in chemotactic response to diacetyl between control and conditioned worms within strains. ns, non-significant, * *p* < 0.05, *** *p* < 0.001. All bar graph data are reported as mean ± standard error of the mean (SEM). The genotypes shown in the key in panel B are the same for panel C.

**Figure 6 cells-13-01598-f006:**
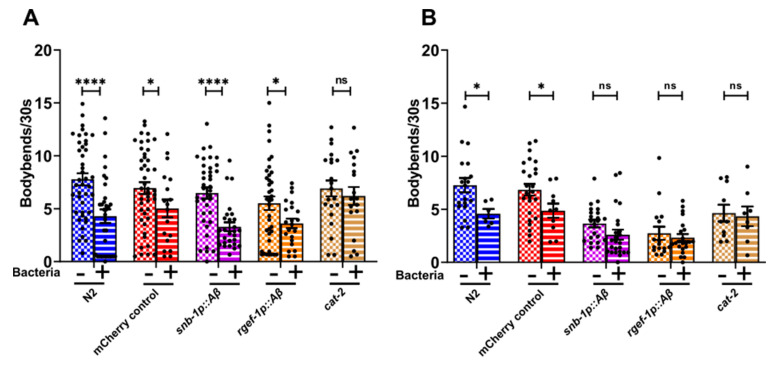
Pan-neuronal *Aβ*-expressing strains show diminished basal slowing response mediated by dopamine. (**A**) day 4. (**B**) Day 8. (n = 3, 5–15 worms/replicate). An unpaired t-test was used to test for significant differences in worm movement on seeded and unseeded plates. ns, not significant, * *p* < 0.05, **** *p* < 0.0001. All bar graph data are reported as mean ± standard error of the mean (SEM).

**Figure 7 cells-13-01598-f007:**
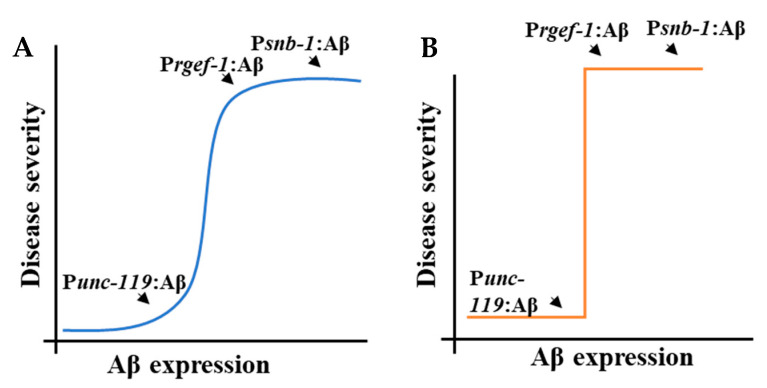
Dose–response curves explaining the correlation between the levels of *Aβ* expression and the severity of disease. (**A**) Gradual dose–response curve. (**B**) Fixed dose–response curve.

**Table 1 cells-13-01598-t001:** Differences in median and maximum lifespan compared to the transgenic control strains.

Strain Name	Median LS (Mean ± SEM)	% Change	Maximum LS (Mean ± SEM)	% Change
N2 control strain	19.3 ± 0.3		32.3 ± 0.3	NA
*mCherry* control	17.7 ± 0.3	NA	30.7 ± 0.7	NA
*snb-1p::Aβ1-42*	13.2 ± 0.2	−26%	19.3 ± 0.3	−36.9%
*rgef-1p::Aβ1-42*	14 ± 0.6	−21%	22.7 ± 0.7	−26.1%
*YFP* control	16 ± 0.0	NA	28 ± 0.0	NA
*unc-119p::Aβ1-42*	15 ± 0.0	−6.25%	22 ± 0.0	−21%

NA: Not Applicable.

**Table 2 cells-13-01598-t002:** Gompertz analysis based on the mortality rate of the *Aβ*-expressing strains compared to the transgenic controls.

Strain Name	Initial Mortality Rate (A)	Gompertz Value (G)	Rate of Aging (MRDT)
N2	3.65 × 10^3^	0.1667	4.15
*mCherry* control	6.00 × 10^3^	0.1677	4.13
*snb-1p::Aβ1-42*	3.89 × 10^3^	0.3217 **	2.16
*rgef-1p::Aβ1-42*	1.91 × 10^3^	0.3881 *	1.79
*YFP* control	3.06 × 10^3^	0.2274	3.05
*unc-119p::Aβ1-42*	3.95 × 10^3^	0.2602	2.66

* *p* < 0.05, ** *p* < 0.01.

**Table 3 cells-13-01598-t003:** Correlation between expression levels and severity of disease phenotype.

	*snb-1p:: Aβ1-42*	*rgef-1p:: Aβ1-42*	*unc-119p:: Aβ1-42*
Expression level	High	Medium	Low
Lifespan	3	2	1
Brood size	3	2	1
Egg retention	3	2	1
Chemotaxis (diacetyl)	3	2	1
Motility on solid media	3	3	1
Motility in liquid media	3	2	1
Basal slowing response	3	3	NA

The difference in severity across the various phenotypes is combined with Aβ expression levels in this table to develop an overall phenotype score. Phenotypic severity, combined with expression levels, is color-coded as follows: red for severe, yellow for moderate, and blue for mild. NA: Not Applicable.

## Data Availability

Data are contained within the article and Appendix A.

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
