# Peer review of "Levels of Amyloid Beta (Aβ) Expression in the Caenorhabditis elegans Neurons Influence the Onset and Severity of Neuronally Mediated Phenotypes"

_cells, 2024, doi:10.3390/cells13181598_

Round 1
Reviewer 1 Report
Comments and Suggestions for Authors
In this manuscript by Sirwani et al, the authors compare the consequences of expression of the human Ab1-42 peptide by three different pan-neuronal promoters in C. elegans. Unlike many of previous reports where groups have over-expressed the putative pathogenic peptide in C. elegans, this group does compare the approximate number of integrated copies (by qPCR) and the relative expression levels from these integrated copies (by qRT-PCR) to attempt to determine the link between expression levels and potential pathological consequences. The group divides the lines into high, moderate and low, and then completes a suite of analyses that examine lifespan, behavioral function, and healthspan-related outcomes. The amount of work that has gone into this is to be commended, but there are some significant concerns that should be addressed before the work could be considered acceptable for publication (please see below). It should be noted that the pdf copy sent for review had a number of either conversion or other errors that detracted from this reviewers enthusiasm for doing the review. This includes a cropped figure that appears on Page 1 (appears to be the top of a Figure that does not appear in the manuscript), several “Error! Reference source not found” statements, and inclusion of the comments window from Word.
Major Concerns
1) The authors use two newly created and integrated strains, whereby the human Ab1-42 peptide is expressed under the control of the snb-1 or rgef-1 promoter, as well as an existing line where it is expressed under the control of the unc-119 promoter. The control for these are a Pmyo-2::RFP or a Pmyo-2::YFP construct, because those were (presumably) the respective co-injection markers. These are not appropriate controls because they do not include another plasmid that has multiple copies of either the snb-1 or rgef-1 promoter generated in the same fashion. A more stringent set of controls, to examine the consequences of Ab1-42 would be Ab1-40 and/or another potentially aggregating protein. However, at the very least an attempt to control for the possible consequences of additional copies of promoters should be included. Promoter usage, and the subsequent competition with any endogenous genes could significantly affect the results. This is particularly concerning given the relatively similar levels of expression from the rgef-1 lines and snb-1 lines, but quite different behavioral outcomes (in some cases).
2) As mentioned below it is not clear whether the lines being examined are extrachromosomal or integrated, but more concerning is the apparent use of only a single line for these. A more standard approach is to use three independently isolated lines, although there are examples where investigators use two. It would be completely unwieldy to complete the
3) The authors have compared 3 different Ab1-42 expressing lines, ostensibly to compare expression levels as a potential variable. But, in the majority of the comparisons the experimental lines are only unpaired t-tests between the individual lines and the control. ANOVAs are used for the age-related changes where age is a variable. If concentration is being tested, then the authors need to use an ANOVA to in which expression level is included as a variable.
4) The authors have categorized their different lines as high, moderate and low, although they are distributed in that manner. The differences in expression level between the Psnb-1 lines and Prgef-1 lines are moderate at best. Further, it is unclear when the expression levels were determined. Are those all day 1 animals? I think this is a critical point in the rationale for the study, that is, that different expression levels are contributing to different phenotypic outcomes. One could imagine that the expression levels of the transgenes diverge during aging, which would be an important point. In addition, even if the endogenous locus is expressed at a given frequency (and the authors provide a good faith effort to explain this), there are any number of post-transcriptional mechanisms that might contribute to the different FPKMs of snb-1 and rgef-1 mRNAs that are independent of the promoter and might not apply to the heterologous gene being expressed. The authors should include those factors in their discussion of expression levels.
Minor Concerns
1) It is not clear if the transgenic strains created here are being maintained as extrachromosomal arrays or are integrated into the genome. I am assuming they are arrays, but that should be clarified. Comparing arrays, which as the authors mention later, can be mosaic within animals and between generations is complicated, especially in comparing to integrated strains which can be influenced by local factors.
2) There is a comment “For GFP transgenic strains the progeny was [sic] selected based on pan-neuronal GFP expression.” No pan-neuronal gfp plasmids were described in the methods or the strains constructed.
3) In the introduction, the authors use the term “Imprecise” to categorize gamma-secretase cleavage. I am not sure that is the best word to use here. The initial cleave site can vary as one of two sites, but then processive cleavage of APP is precise in a tripeptide fashion. I would encourage the authors to consider a different word.
4) In table 1 the median lifespans are calculated to 2 significant digits which would reflect an accuracy of less than 2 hours, given that these animals were moved every 1-2 days, I would encourage the authors to report to one decimal place. Also, please confirm that there was no observed variation (+/- 0.00) for the last two lines entered, that precision seems statistically unlikely.
5) The discussion of the dopaminergic phenotypes should be qualified with other potential sensory issues, which may be being interpreted as dopaminergic. Without specific testing of the dopaminergic neurons, I would strongly recommend the authors report the behavior without the specific conclusion that the differences are dopamine-mediated.
Suggestions
1) There are a few grammatical errors or oversights that have occurred. For example, in line 467, there is a “this” that should not be capitalized. I would encourage additional proofreading throughout the manuscript.
Comments on the Quality of English Language
The English is quite good overall. There are a few minor typographical errors that are easily corrected.
Reviewer 2 Report
Comments and Suggestions for Authors
In this study, Sirwani et al., generate two new pan-neuronally expressed AB strains as a model to study AD in C. elegans. They measured copy number, expression level and phenotypes in three strains, one of which has been previously published. They find that worms expressing snb-1p::AB exhibit decreased lifespan and heathspan phenotypes, and suggest that these worms would be a better tool to use for future AD studies than a previously published strain (unc-119p::AB). I feel with some changes to the writing, increased controls and increased biological replicates, a heavily revised manuscript could become suitable for publication.
I think it is a bit strange that the authors chose to refer to strain names versus assigning integrated array numbers (eg. since the lab’s codes are WG/au, it would be auIs1234 etc.). I am also ambivalent about their transgenics as the authors make no mention of multiple lines of each of their transgenics. Was only one line established per injected construct? How did they determine which was a suitable line to use?
Line 212: italicize cdc-42. Also multiple instances where promoters and C. elegans were not italicized.
Line 215: Reference problem here, and also throughout.
I would be careful about comparing copy number to expression. I think this is nicely exemplified by the integrated unc-119p::AB strain possessing so few copies (3.59) and its relative qPCR expression was 5.2, whereas the strains the authors generated had 75+ “copies” and only a relative expression of 11.67-13.47.
Line 374: please provide a reference describing rgef-1 as a weak promoter. I wasn’t aware that it is considered a weak promoter and it’s one of the most popular pan-neuronal promoters. Many people have made single copy insertion transgenes using the rgef-1 promoter and in some cases it expresses better than an rgef-1 integrated transgene because it’s not in a complex array. In the discussion the authors discuss the FPKM of the two different genes (snb-1 vs. rgef-1), but I think that at the end of the day, these things are being expressed in complex arrays and it is hard to compare.
It is curious that the authors chose to use only a strain expressing myo-2p::mCherry as a control for their strains. I think a more appropriate control would be to have cloned GFP behind the snb-1 and rgef-1 promoters and injected that with the myo-2 plasmid. This is a nice way to show that expressing the AB protein does something, whereas injecting just rgef-1p::gfp does little (in reality it probably does have a minor effect – lifespan is such a delicate phenotype). It would also be a nice way to show that there is no toxicity elicited by just the promoter dosing alone – for example, expressing anything (even a myr-GFP) under the unc-86p at anything over 1ng/ul can cause axon guidance defects in HSN and results in a pretty strong egg-laying defect.
Table 1/Figure 2: I am a bit perplexed by the unc-119p::AB experiments here. It is obvious to me by both Table 1 and Figure 2 that the authors did not perform multiple biological replicates of the YFP control and unc-119p::AB (SEM is 0 for both median and maximum). I am also confused by the calculations that they did. Is the % change for the median lifespan only -6.25%? It looks way more than that. Assuming that the graph shown in Figure 2A has the only biological replicate of the unc-119p::AB experiment, the maximum lifespan of the YFP control is something like 27 or 28? Why is it reported as 22 in Table 1? I think the median lifespan numbers might be off too. Figure 2B is not helpful and just rehashes Table 1. If the goal is to introduce p-values, it can be added to Table 1.
Figure 3: The graph organization is simply disorganized. Please make all graphs comparable sizes and be consistent regarding bar widths used for the statistic comparisons. It is intriguing that expression of the AB protein in the nervous system would actually affect the worm’s fecundity, is this consistent with previous reports/has this been thoroughly investigated? If yes, please provide citations. I wonder if this could be an effect of the unc-54 3’UTR that was used in all the constructs. Usage of this UTR does get expression where you don’t always want it, predominantly in muscle. For future work I would suggest to switch to something like the let-858 3’UTR. Figure 3D – on Day 8 between the YFP and unc-119p::AB experiments – could you provide the statistical test? It looks like it might be significant. For Day 12, although it’s not significant, it seems like the rgef-1::AB worms move even faster than the control, although maybe not significant, it is a trend worth commenting on.
Figure 4: Same disorganization problems as Figure 3. If anything, adding all those comparisons, including the not significant ones is jarring compared to Figure 3. I think you need to pick one way – show everything (as done in Figure 4) or figure out a more elegant way to show the comparisons or select comparisons to show (as done in Figure 3). The spacing between the graphs is unflattering (ie. Panel D’s comparisons impinging on Panel B).
Lines 552-553: I am very confused by the arguments the authors are making here. I do not understand how they are making this conclusion, is this from Supplemental Figure 2B? The authors explain the parameters well, but should explain that a higher/larger/shorter phenotype means XYZ. Because the unc-119p::AB worms curled less on Day 12 that somehow means that it did not show rapid changes in any of the parameters (you didn’t see any difference in the rgef-1 and snb-1 strains too right? At least no significant tests done on the graph)? Just by that one significant comparison the authors make a strong claim in lines 553-555.
Supplemental Figure 1: No reference in the text. Not sure why this was included?
Figure 5: It’s distracting that the colour scheme used in panel D is so different. Also, the naming convention went back to using strains as opposed to what is actually the genotype of the worms. Why was the decision to use N2 as a control only used here? With all worm aging studies it is the convention to use N2 as a baseline control, even if you have transgenic animals as the ‘true’ control. It is a great way for other worm aging labs to see how you handled the worms – that is, if your N2 experiments are consistent with the plethora of other studies, it is easier to appreciate manipulations where you do see effects. I would like to see the authors include an N2 control for all their experiments, as is the standard in the field.
Figure 6: Please repeat these experiments, particularly more of the YFP control, because the data, as the authors state plainly is ‘preliminary’ in Lines 638-640, is simply not convincing. Since these transgenes are expressed pan-neuronally, it would be curious to see more the global effects of neuronal function being affected. One could perform a simple aldicarb assay to see if synaptic transmission is impaired.
Discussion:
Line 746: “Mutants with decreased brood size have low levels of dopamine”. This sentence is problematic and is not universally true. As far as I know, low levels of dopamine just makes the worms lay less eggs. There are certainly genes that have mutants with low brood sizes but have nothing to do with dopaminergic neurons.
Line 795: Kinking instead of kinky.
Inconsistent usage of citations, in the discussion there’s the numbered citations and then randomly Name et al., Year.
Figure 7: I feel that the authors are doing a disservice to the promoters by separating the promoters’ ability in terms of disease phenotype as they show. The authors did not do any attempts to titrate the amount of copy numbers of the individual constructs and provide only a single line for each transgene. Is it fair to say that rgef-1 is not as good as snb-1 based on a single strain? Probably not.
The authors should frame it more that they built some strains, and that the newer strains might be more useful than the older strain. It is disappointing that they do not suggest any sort of titration experiment in the future to see if this holds true but mention that single copy insertions/CRISPR might be a better option to control copy numbers.
Round 2
Reviewer 1 Report
Comments and Suggestions for Authors
I have no additional comments, the authors have satisfactorily addressed my concerns.
Comments on the Quality of English LanguageNone
Author Response
Thank you for your positive feedback and for acknowledging the revisions made to address your concerns. We appreciate your thorough review and are pleased that the changes have been satisfactory.
Reviewer 2 Report
Comments and Suggestions for Authors
The authors have made significant changes to their revised manuscript and I have made an effort to introduce additional controls that I have suggested. However, I feel there are still quite a few things that must be done before this article can be accepted. Importantly, the quality of the material presented is still lacking. The presentation of graphs and micrographs feels rushed and makes it distracting and frustrating for the reader.
Figure 1B – is it possible to use a Box and Whiskers plot instead of a bar graph here, or at least show your replicates?
Figure 1C is not helping that much presented as a bar graph. If you must include it, consider moving it to the Supplementary Figures section or maybe even consider making it into just a single sentence.
This is more of a stylistic point, but it’s more common in scientific writing to write results statements, followed by a reference to the figure. For example, the authors have written:
As can be seen in Supplementary Figure A1C, the GFP expression is evident in the head and tail neurons in addition to the entire neuronal network throughout the length of the worm.
It should rather be phrased as:
“We observed prominent pan-neuronal expression of GFP in both of our imaging experiments, consistent with previous reports characterizing snb-1 and rgef-1 expression patterns (Supplementary Figure A1C) [refs].”
I am unclear why the authors used the term “neuronal network” – this is not what you think it means in this context.
The statement “The GFP expressing spots on the body of the worm indicate cell bodies.” comes across as lazy. What are these cell bodies? If you don’t want to determine their identity, that is fine, and you do not need to bring it up (but if you’re curious, my best bet is that they’re probably mechanosensory neurons). I would just take out that statement since it doesn’t add anything.
“Survival curves of the transgenic C. elegans strains described in section 3.1 are shown in Figure 2. The Mantel-Cox (log rank) test was used to determine the difference in the distribution of these survival curves. Data on lifespan experiments for all biological replicates are summarized in Error! Reference source not found..”
This should not be in the main text, but rather in the materials and methods. This clogs up your results section.
The next step was to evaluate whether variation in the expression of Aβ1-42 influenced the severity of healthspan indicators such as fecundity and maximum speed.
This was written too casually and not giving enough framing. It should be something along the lines of “We sought to test if…” versus “The next step was”
Figure 3:
Graph D is offset from Graph C. The ordering of the genotypes in the legend is off; move N2 to the top. Why were unpaired t-tests used here instead of ANOVA? The authors use ANOVA in other graphs.
The authors did a nice job explaining egg laying, but there should be more background leading into the in utero egg retention section – maybe just one sentence with an appropriate reference.
Figure 4:
The placement of the asterisks (also sometimes bolded?) is chaotic and inconsistent. Consider adjusting the spacing on the graphs. Check to see if the font sizes are all consistent.
For Figure 4 and 5, it might be useful to put in the figure legend that the strain Legend/key (for genotypes) is the same across multiple panels, as it’s the same genotypes in 4a-d right?
Furthermore, the Aβ1-42-expressing strain unc119p::Aβ also shows a significant reduction in the chemotactic ability towards diacetyl in middle-aged worms on day 8 (0.50 ± 0.082) compared to the transgenic control GRU101 (0.77 ± 0.07) (p = 0.045, unpaired t-test), but not on day 4 (p = 0.36).
Change GRU101 to have the genotype instead of the strain name.
Figure 5
The colour scheme used in these graphs is not consistent with the rest of the text. Consider changing panel D to make it less visually jarring. The size of the text in D is miniscule and impossible to read.
Figure Legend: for B and C, Why were t-tests used here and not ANOVA? There is no descriptor in the legend for D.
Discussion
The discussion is not a place where we report results, it should be meant for actual discussion. The first time Supplementary Figure A3 was mentioned was here (also mentioned in the text as ‘Supplementary A3’, missing ‘Figure’), which is inappropriate. The insertion of this text belongs in the Results section:
“Moreover, to further validate the severity of the phenotypes observed in our human Aβ-expressing strains, we generated a non-toxic mouse Aβ1-42 transgene expressing strain driven by same pan-neuronal promoter snb-1. The mouse Aβ1-42 peptide, while capable of aggregation, is significantly less prone to form the toxic oligomers characteristic of the human Aβ1-42 peptide. The mouse Aβ-expressing strain does not exhibit severe behavioral defects in lifespan, brood size, or motility when compared to the human Aβ1-42 expressing strain or the transgenic mCherry control strain (Supplementary A3). This suggests that the toxic phenotypes we observe are indeed specific to the human Aβ1-42 peptide, rather general consequence of expressing any aggregating protein or the result of promoter competition.”
In the Discussion you should come back to this, and maybe have some references about the mouse peptides?
Appendix A:
The appendix is an optional section that can contain details and data supplemental to the main text—for example, explanations of experimental details that would disrupt the flow of the main text but nonetheless remain crucial to understanding and reproducing the research shown; figures of replicates for experiments of which representative data is shown in the main text can be added here if brief, or as Supplementary data. Mathematical proofs of results not central to the paper can be added as an appendix.
Authors left in the instructions, please remove them.
Supplementary Figure A1:
Transgenic C. elegans strain showing pan-neuronal GFP expression driven by snb-1 promoter fragment. GFP expression is evident in the head and tail neuron regions. The GFP expression on the body of the worm indicates the network of nerves running along the length of the nervous system (Scale bar = 50 μm).
I think to the average reader this is not obvious. Since the authors put in the effort to label the following image in D, why not also in C? I would like to see the authors mention what stage these worms are. The size of the worms in panel C and D are totally different – it seems like it’s a young L2 animal in C and maybe an L4 or young adult animal in D. I would also appreciate a scale bar in panel D that is more obvious. The authors are not consistent in between panel C and D with regards to labelling. I am concerned that the authors have not written any details regarding microscopy in the Materials and Methods section (what microscope did you use? What objectives? What camera? How did you mount the worms (did you use agarose pads? What anaesthetic did you use? What stages did you image?).
It is also frustrating that the images appear to show the worm quite out of focus, but if this experiment cannot be repeated, it will suffice.
Please change the colour of the text of the labels in D to white to make it easier for the reader. I am also confused about your arrowheads. They are pointing downward. Also, you need to be more specific about the nerve cords. What we are seeing is bright expression in the ventral nerve cord, or you can say you see expression in both nerve cords. In the main text, image and figure legend, change NC to be VNC, as is the convention in worms.
The “T” for ‘tail’ label is so far away from the worm that it’s just pointing to a black space. There also seems to be a bunch of random artifacts on the image – black lines etc.
Typo: missing a dash in “An unpaired t test”
Supplementary Figure A2:
Labelling N2 as “N2 wt type” is redundant. Change to N2. This comment is relevant across the text.
Supplementary Figure A3:
Check sizing of graphs. The dots in C are so much larger than in D, while the graph in D is actually much larger than in C (very obvious, looking at the Y axis between the two graphs) Graphs are not horizontally aligned, making it frustrating to the reader.
Minor Edits:
Figure Legends – you should say what every bar means. Does the bar represent medians, and the error bars represent standard error?
The genomic DNA was digested with the 4-base cutting restriction enzyme, Sau3A (New England Biolabs, Inc.).
I think it should be Sau3AI right? It would be nice if you included your conditions for preparing the digestion of this genomic DNA and inactivation of the enzyme. Alternatively, provide a reference describing this protocol.
As a control for appropriate pan-neuronal expression, the same snb1 promoter fragment was also inserted into the complementary sites XbaI (5’) and XmaI (3’) of the promoter-less GFP containing plasmid, pPD95.77 (Fire vector kit 1995) [29], resulting in pan-neuronal GFP expression plasmid pSNB1GFP.
Because of the style chosen in the rest of the text, please introduce a dash (-) to make the plasmid name pSNB-1GFP.
Typos throughout:
ex: “we have labeled strains using their promotor and Aβ status: the standard Bristol N2 strain as wild-type N2 strain”
Correct to 'promoter'
Spacing issues: ex; “GRU102 strain asunc119p::Aβ.” Needs a space between ‘as’ and unc-119p”
Inconsistent usage of Commonwealth English and American English, notably “behavior” and “behaviour” are both used.
Multiple references not found: Error! Reference source not found.
When reproduction is affected adversely, physiological decline starts early because reproduction in C. elegans hermaphrodites is completed while the animal is relatively young, whereas most of what lifespan assays measure is post-reproductive.
This sentence is awkward, consider rewriting it.
To measure egg-laying rate in these transgenic strains, egg-laying assays were performed.
This is trite. You could say something like “We sought to measure […] in […] in order to determine the effect of […] on reproduction”.
Comments on the Quality of English Language
The English is mostly fine. There are a few typos littered throughout the text.
Round 3
Reviewer 2 Report
Comments and Suggestions for Authors
The authors have done a great job addressing my concerns regarding presentation. The figures are significantly improved.